# DEMO: Diffusion-based Evolutionary Optimization for 3D Multi-Objective Molecular Generation

## Abstract

Optimizing multiple objective properties while satisfying structural constraints is a major challenge in 3D molecular discovery. This difficulty arises because optimization objectives can be non-differentiable and the structure–property relationship is often unknown. Evolutionary algorithms (EAs) are widely used for multi-objective optimization to find Pareto fronts and can naturally handle structural constraints without any explicit modelling; however, in the 3D molecular space they lack mechanisms to guarantee chemical validity and are therefore prone to producing invalid structures. Conversely, diffusion models excel at generating chemically valid 3D molecules but typically require modifying the model and retraining to incorporate structural constraints. Moreover, diffusion models are not inherently designed for direct multi-objective optimization and struggle to explore the Pareto front of the learned property distribution — a critical capability for discovering novel, high-performing molecules. To bridge this gap, we propose a novel 3D molecular multi-objective evolutionary algorithm that leverages the generative power of a pretrained diffusion model. Instead of manipulating molecules directly in the complex chemical space, our method performs crossover operations in the noise space defined by the diffusion model's forward process, thereby enabling parental features or desired fragments to be fused into offspring. The pretrained model's denoising process then restores structural validity. The approach is highly composable and, requiring no retraining, can be readily integrated with existing guidance methods to improve discovery. Experimental results demonstrate strong performance on single-objective, multi-objective, and structurally constrained optimization tasks. Notably, our hybrid method successfully and rapidly explores and captures the Pareto front of the learned property distribution, effectively overcoming a key limitation of using diffusion models alone.

## 1 Introduction

Molecular discovery is a core part of the early stages of materials science and drug design Gong et al. (2024a). Traditional molecular discovery processes, whether through high-throughput screening or empirical expert design, are extremely time-consuming and labor-intensive, and are unable to effectively explore the vast chemical space. In recent years, efficient computational generation methods Ho et al. (2020) have significantly accelerated this process, enabling the effective discovery of novel molecules with desired properties.

Discovering novel molecules typically requires either finding molecules with given properties (reverse design) or simultaneously optimizing multiple objectives of lead compounds (lead optimization) Fromer & Coley (2023), and sometimes enforcing specific structural constraints Ghorbani et al. (2023). For instance, a drug discovery aims to simultaneously optimize a lead compound for maximum potency and minimum toxicity, while being structurally constrained to retain a specific chemical scaffold essential for binding. A variety of molecular generation methods, based on search Zhou et al. (2019c) and learning Beaudoin et al. (2024), have been proposed to design desired molecules. However, most focus on 1D SMILES Gong et al. (2024a) strings or 2D molecular graphs Wang et al. (2025). While these representations are computationally lightweight, they lack true 3D geometry and stereochemical information, limiting their ability to capture chirality, conformational

diversity, and spatial complementarity. This deficiency hampers accurate modeling of intermolecular interactions, property prediction, and downstream molecular dynamics or quantum chemistry simulations. In contrast, 3D representations provide a faithful spatial description of molecular conformations, including bond lengths, angles, torsions, stereochemistry, and noncovalent interactions, making direct exploration of 3D molecular space a superior strategy Thomas et al. (2018).

Diffusion models Hoogeboom et al. (2022); Xu et al. (2023) excel at generating high-quality, diverse 3D molecules by using a reverse denoising process to enforce chemical validity. Nonetheless, they face significant challenges in molecular discovery. First and foremost, they are ill-suited for multi-objective optimization because they operate as rigid conditional generators, not flexible optimizers. This core limitation means that simultaneously steering generation toward several, often conflicting, properties is inefficient. Jain et al. (2023); Fetaya et al. (2019) Secondly, the models lack flexibility, as adapting them to a new property or a structural constraint is computationally prohibitive, which often requires extensive fine-tuning or even a complete model redesign to manage the complex coupling between structure and properties Peng et al. (2023).

Evolutionary algorithms Holland (1992) (EAs) are widely used in molecular discovery. Without any training, their population-based optimization is naturally suited for handling both multi-objective optimization and constraint satisfaction Deb et al. (2002). Moreover, EAs are gradient-free black-box optimizers, enabling direct use of molecular evaluators commonly found in open-source software, for which gradients are typically unobtainable. However, extending traditional EAs to the 3D molecular space is nontrivial, as their genetic operations on atomic coordinates are unaware of chemical laws, frequently generating invalid structures and thus impeding search efficiency.

In this work, we introduce Diffusion-based Evolutionary Molecular Optimization (DEMO), which integrates diffusion models into evolutionary algorithms to improve molecular validity and accelerate optimization. DEMO exploits the forward diffusion process to temporarily hide complex chemical and geometric constraints of 3D molecules while preserving essential information. Crossover operations are then conducted in the noise space defined by the forward process. Finally, the reverse diffusion process reinstates the 3D chemical constraints to generate chemically valid structures. Our contributions are as follows:

1. **To resolve the validity crisis of EAs in 3D space:** We introduce a novel noise-space crossover operator. By manipulating molecular representations in the diffusion model's latent space and leveraging the denoising process to ensure validity, our method enables robust genetic operations without sacrificing chemical correctness.

2. **To overcome the optimization and flexibility limits of diffusion models:** We embed a pretrained diffusion model within an EA framework. This turns the rigid generative model into a flexible, gradient-free black-box optimizer, making it adept at multi-objective optimization and capturing the Pareto front—tasks that are notoriously difficult for standalone diffusion models—without any need for retraining.

3. **To provide a unified and powerful optimization framework:** We present DEMO, the first framework to successfully bridge EAs and 3D diffusion models. We demonstrate its state-of-the-art performance across a comprehensive suite of benchmarks, from property targeting to constrained Pareto optimization, establishing a new and powerful paradigm for automated molecular discovery.

## 2 BACKGROUND

**Problem Formulation** Finding molecular structures with multiple desired properties can be framed as **single-objective optimization (SOP)** or its constrained variant **(CSOP)**. This approach scalarizes multiple property goals into a single objective, typically by minimizing a distance metric $d$:

$$\min_{M \in S} d((f_1(M), \dots, f_S(M)), y^*) \quad \text{s.t.} \quad C(M) \tag{1}$$

where the property vector for a molecule $M$ is composed of outputs from $S$ individual predictors $(f_1, \dots, f_S)$, $y^*$ is the target vector, and $C(M)$ represents structural or property-based constraints.

In contrast, multi-property optimization based on lead molecules optimization can be framed as **multi-objective optimization problem (MOP)**, or a **constrained MOP (CMOP)**, handles multiple, potentially conflicting objectives simultaneously without scalarizing them Zhou et al. (2019a;b).

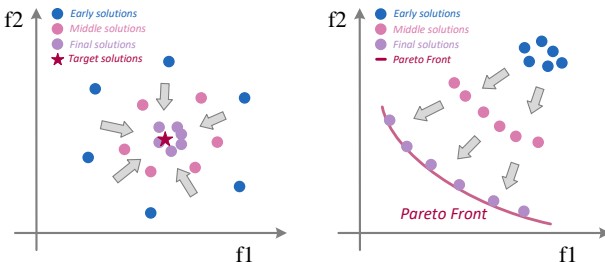

Figure 1: Molecular discovery include: Left: finding a molecular structure with desired properties (SOP); Right: optimizing multiple properties simultaneously (MOP).

The goal is to find a set of optimal trade-off solutions, known as the *Pareto front (PF)*. A solution is *Pareto optimal* if no single objective can be improved without degrading at least one other. This is determined by *Pareto dominance*: for minimization, $M_1 \succ M_2$ if it is superior in at least one objective and not inferior in any:

$$\forall k \in \{1..K\} : f_k(M_1) \leq f_k(M_2)$$
$$\wedge \quad \exists k \in \{1..K\} : f_k(M_1) < f_k(M_2) \tag{2}$$

The set of all non-dominated solutions constitutes the PF, which is the desired output of the MOP:

$$\min_{M \in S} F(M) = (f_1(M), \ldots, f_k(M)) \tag{3}$$

A **constrained MOP (CMOP)** extends this by incorporating constraints. The desired output is the constrained PF, which comprises all non-dominated solutions that also satisfy $C(M)$.

**3D Molecule Representation** Molecular structures in 3D space are typically represented as a tuple $M = (X, H)$, where $X = (x_1, \ldots, x_n) \in \mathbb{R}^{3 \times n}$ denotes the 3D coordinates of $n$ atoms, and $H = (h_1, \ldots, h_n) \in \mathbb{R}^{a \times n}$ encodes $a$ atomic features. A fundamental property of molecular systems is their invariance to rigid transformations of $X$, while the generation process must be equivariant to these transformations. Formally, for a rotation/reflection matrix $R \in \mathbb{R}^{3 \times 3}$ and translation $t \in \mathbb{R}^3$, invariance implies: $f(RX + t, H) = f(X, H)$, where $f$ is a scalar function. Equivariance requires: $g(RX + t, H) = Rg(X, H) + t$, where $g$ outputs 3D coordinates.

**Traditional and Generative Approaches.** While Evolutionary Algorithms (EAs) have been applied to 1D SMILES Xia et al. (2024) and 2D molecular graphs Yu et al. (2024); Jensen (2019), they are fundamentally hampered by their inability to model 3D geometries, leading to invalid structures or a failure to capture stereochemistry. Conversely, modern generative models (e.g., Flow Jin et al. (2025), VAEs Gong et al. (2024b), Diffusion Morehead & Cheng (2024)) excel at learning complex 3D molecular distributions. However, their primary drawback is inflexibility; conditioning these models on new properties or constraints typically requires costly retraining, making them ill-suited for rapid, iterative optimization tasks.

**3D Diffusion Models and Guidance Limitations.** 3D diffusion models are particularly effective, generating high-quality molecules by learning to reverse a forward noising process, $q(M_t \mid M_{t-1})$, with a denoising network, $p_\theta(M_{t-1} \mid M_t)$. However, steering their generation toward desired properties remains a major challenge. Existing guidance mechanisms either necessitate expensive retraining or fine-tuning for new tasks (e.g., classifier-based guidance Dhariwal & Nichol (2021); Ho & Salimans (2022)) or, if training-free, are often inefficient, restricted to differentiable objectives, and struggle to balance multiple properties or complex structural constraints without degrading sample quality Ye et al. (2024).

## 3 METHOD

### 3.1 OVERVIEW OF DEMO

DEMO implements a diffusion-based evolutionary loop that combines a pretrained 3D diffusion model with classical evolutionary operators. The algorithm initializes a population of size $N$ by

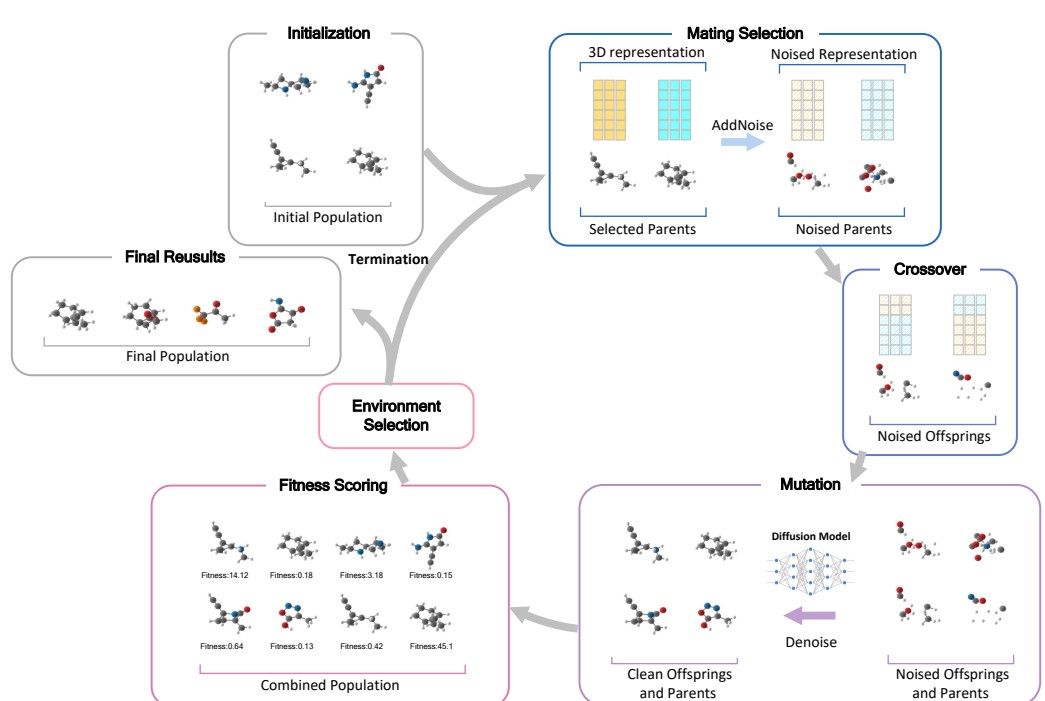

Figure 2: DEMO combines EA with Diffusion models to iteratively optimize 3D molecular.

sampling from a database or by guided sampling from $p_\theta$ with guidance $\mathcal{M}$. The fitness is computed (details in Sec.3.3) with respect to the task specification $C$ (a target vector) and a 3D fragment $F$ to be included in the target molecule.

DEMO runs the following loop each generation: parents are selected by binary tournament selection and split into crossover and mutation pools. The sizes of these pools are determined by a linear schedule, which favors crossover operations in the early stages and shifts towards mutation operations in the later stages. The crossover operator (details in Sec.3.2) facilitates global exploration by first injecting noise of level $t'$ into the parents, then randomly crossing over their noisy representations (both molecular coordinates and features) or those of a parent and the fragment $F$, and finally denoising the result. In contrast, the mutation operator enables local exploration by adding noise to a parent molecule and subsequently denoising it. The noise level $t'$ is annealed from $t_{\max}$ down to $t_{\min}$ as the generation progresses; larger noise levels in the early stages promote broad exploration, while smaller levels later on facilitate fine-grained exploitation. After the candidate molecules are generated, their fitness is computed, and environmental selection is used to choose the top $N$ individuals for the next generation. This entire loop continues for $G$ generations or until another termination condition is met. DEMO ultimately yields an optimized molecular framework that balances fragment fusion, exploration, and exploitation. Crucially, DEMO functions as a black-box optimizer and requires no additional training.

## 3.2 CROSSOVER IN DEMO

For any $0 < t' < T$, the forward diffusion process produces a noised molecular representation $M_{t'}$ from a clean molecule $M_0$:

$$M_{t'} = (X_{t'}, H_{t'}) \sim \mathcal{N}\big(\sqrt{\bar{\alpha}_{t'}}\, M_0,\, (1 - \bar{\alpha}_{t'})I\big). \tag{4}$$

To combine two noised parents, $M_{t'}^{(1)}$ and $M_{t'}^{(2)}$, we perform a crossover operation in the noise space. Since the ordering of atoms in a 3D molecular representation is arbitrary, a simple one-point crossover on the sequence of atoms is equivalent to a random partitioning. This operation swaps segments of both coordinates $(X_{t'})$ and features $(H_{t'})$ between the parents. This recombination is formally a linear combination, expressed using a binary mask $B$ (more details are in Appendix A.1.2),

---

**Algorithm 1** DEMO:Diffusion-based Evolutionary 3D Multi-Objective Molecular Optimization

---

**Require:** Pretrained diffusion model $p_\theta$, population size $N$, generations $G$, noise bounds $t_{\min}, t_{\max}$, guidance $\mathcal{M}$, task $C$, fragment $F$
**Ensure:** Optimized population $P$
1: $P \leftarrow$ sample $N$ molecules from database or guided sampling $p_\theta$ by $\mathcal{M}$ $\quad \triangleright$ initialize population
2: GetFitness$(P, C, F)$ $\qquad\qquad\qquad\qquad\qquad\qquad\qquad\qquad \triangleright$ get fitness according to C and F
3: **for** $g = 1$ **to** $G$ **do**
4: $\quad t' \leftarrow \lfloor t_{\max} - \frac{g}{G}(t_{\max} - t_{\min}) \rfloor$
5: $\quad N_{mt} \leftarrow \lfloor \frac{g}{G} \rfloor N$, set $N_{co} \leftarrow N - N_{mt}$ $\qquad\qquad\qquad\qquad\qquad\qquad \triangleright$ linear schedule
6: $\quad P_0^{co} \leftarrow$ select $N_{co}$ parents from $P$ $\qquad\qquad\qquad\qquad \triangleright$ parent selection for crossover
7: $\quad P_0^{mt} \leftarrow$ select $N_{mt}$ parents from $P$ $\qquad\qquad\qquad\qquad \triangleright$ parent selection for mutation
8: $\quad P_{t'}^{mt} \sim q(\cdot \mid P_0^{mt}), \quad P_{t'}^{co} \sim q(\cdot \mid P_0^{co}), \quad F_{t'} \sim q(\cdot \mid F)$ $\qquad\qquad \triangleright$ add noise
9: $\quad O_{t'}^{co} \leftarrow$ crossover$(P_{t'}^{co}, N_{co})$ $\qquad\qquad\qquad \triangleright$ produce offspring from parent pairs
10: $\quad O_{t'}^F \leftarrow$ crossover$(P_{t'}^{co}, F_{t'}, \frac{1}{2}N)$ $\triangleright$ produce extra offspring from fragment and parent pairs
11: $\quad O_{t'} \leftarrow P_{t'}^{mt} \cup O_{t'}^{co} \cup O_{t'}^F$ $\qquad\qquad\qquad\qquad\qquad \triangleright$ offspring pool in noise space
12: $\quad$ **for** $s = t', \dots, 1$ **do**
13: $\qquad$ sample $O_{s-1} \sim p_\theta(O_{s-1} \mid O_s, \mathcal{M})$ $\qquad\qquad\qquad\qquad \triangleright$ parallel denoising
14: $\quad$ **end for**
15: $\quad$ GetFitness$(P \cup O_0, C, F)$ $\qquad\qquad\qquad\qquad\qquad\qquad\qquad \triangleright$ get fitness
16: $\quad P \leftarrow$ EnvironmentalSelection$(P \cup O_0, N)$ $\qquad\qquad \triangleright$ keep best $N$ by fitness
17: **end for**
18: **return** $P$

---

to produce the offspring $M_{t'}^{(o)}$:

$$M_{t'}^{(o)} = B \odot M_{t'}^{(1)} + (1 - B) \odot M_{t'}^{(2)}. \tag{5}$$

The key insight is that this offspring, being a linear combination of two samples from Gaussian distributions, is itself a sample from a new Gaussian distribution. While its mean is now a combination of the two original clean parent, it retains the same isotropic noise variance $(1 - \bar{\alpha}_{t'})$. Consequently, the pretrained denoising model can process this novel latent representation and map it to a valid 3D molecule that hybridizes features from both parents. As $t'$ increases, the influence of the combined mean diminishes relative to the large isotropic noise, ensuring the offspring remains in a region of the latent space that the model can readily denoise.

From the manifold perspective, the injected noise "thickens" the molecular manifold into a locally linear region (provided $t'$ is not extremely small), so permuted samples remain within this linearized neighborhood. At the same time, smaller $t'$ values are preferred because they retain more of the original molecular information and require fewer denoising iterations to recover valid $M_0$. Thus, selecting an intermediate $t'$ balances manifold linearity (valid crossover) against information retention (efficient reconstruction), more detail see A.2.

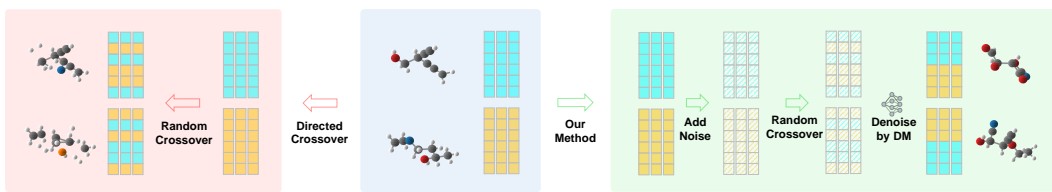

Figure 3: Random crossover on 3D molecules rarely yields valid structures. However, adding Gaussian noise (forward process), applying crossover, and then denoising by a Diffusion model can produce valid molecules.

The choice of the noise level $t'$ is governed by a key geometric condition: the blur radius $(\sigma_{t'})$ must be sufficient to 'flatten' the local curvature of the molecular manifold, characterized by its maximal

radius $R_{\max}$. This implies the relationship:

$$\sigma_{t'} = \sqrt{1 - \alpha_{t'}} \geq R_{\max}. \tag{6}$$

While this provides a formal lower bound, analytically determining $R_{\max}$ for such high-dimensional manifolds is computationally intractable. Therefore, rather than estimating this volatile geometric property, we adopt a more direct and powerful strategy. We treat $t'$ as a crucial hyperparameter and determine its effective range empirically through an offline grid search, optimizing directly for our ultimate objective: the generation of chemically valid molecules.

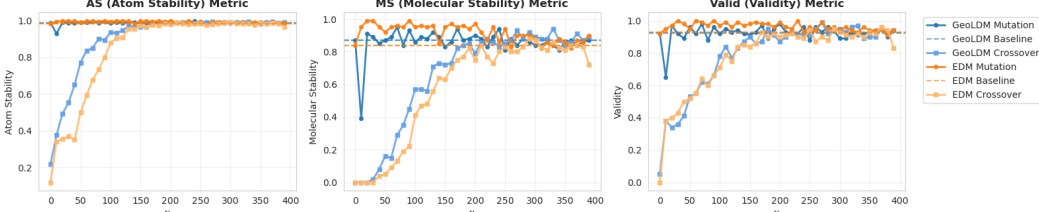

Figure 4: stability, atomic stability, and validity of the samples $M_0^{rec}$ reconstructed from $M_{t'}$

We experimented with EDM and GeoLDM models pre-trained on the QM9 and GEOM-Drugs datasets. Starting from valid molecules ($M_0$), we generated noised parent ($M_{t'}^p$) and offspring ($M_{t'}^o$) molecules at various $t'$ levels. We then evaluated the stability and validity of the reconstructed molecules ($M_0^{rec}$) against the original samples. The results (Figure 4) show that for the QM9 dataset, the quality of molecules reconstructed from offspring ($M_{t'}^o$) is comparable to the original samples when $t' > 200$, and from parents ($M_{t'}^p$) when $t' > 20$. On the GEOM-Drugs dataset, these thresholds increased to 250 and 40, respectively (see A.1.3). This suggests that the choice of $t'$ is largely dependent on the training dataset.

### 3.3 FITNESS EVALUATION

#### 3.3.1 FITNESS DESIGN IN SOP

Finding 3D molecules with target properties $y_i^*$ is formulated as a SOP. Each candidate molecule $M$ is scored by a set of pretrained predictors $f_i(M)$. After normalizing both predictions and targets over the population, the property deviation is calculated as:

$$D(M) = \sqrt{\sum_i \left( \hat{f}_i(M) - \hat{y}_i^* \right)^2}. \tag{7}$$

Constraints are expressed via non-negative penalty functions $c_j(M)$. For a given constraint, we first define an underlying function $g_j(M)$ such that the constraint is satisfied if $g_j(M) \leq 0$. The penalty is then given by: $c_j(M) = \max(0, g_j(M))$. For the structural constraint of fragment presence, we define the penalty as $c_F(M) = 1 - \rho_F(M)$, where $\rho_F(M)$ is the proportion of fragment $F$ within $M$. This ensures that $c_F(M) = 0$ only when the fragment is present at its maximal proportion (we describe how to calculate this in A.1.5). Note that when avoiding fragment $F_{un}$ from appearing in $M$, $c_{un}(M)$ is calculated as $c_{un}(M) = \mathbb{I}\{\rho_F(M) = 1\}$, where $\mathbb{I}\{\cdot\}$ is the indicator function. The total constraint violation (CV) is the sum of all penalties:

$$\text{CV}(M) = \sum_j c_j(M) + \sum c_F(M) + \sum c_{un}(M). \tag{8}$$

We use the Constraint Violation (CV) score to measure a molecule's proximity to the feasible region. Following the $\epsilon$-constrained approach Takahama et al. (2005), we treat solutions as *pseudo-feasible* if $\text{CV}(M) \leq \epsilon$, where the tolerance $\epsilon$ is dynamically adjusted during the search. This strategy allows the algorithm to explore promising solutions near the feasible boundary, thereby preventing premature convergence to potentially suboptimal but strictly feasible regions.

The final fitness function combines the property deviation score with a penalty for infeasibility:

$$\text{Fitness}(M) = D(M) + \mathbb{I}\{\text{CV}(M) > \epsilon\} \cdot P_{\text{infeas}}. \tag{9}$$

Here, the indicator function $\mathbb{I}\{CV(M) > \epsilon\}$ penalizes any molecule whose constraint violation exceeds the current tolerance $\epsilon$ with a constant penalty $P_{\text{infeas}}$.

### 3.3.2 Fitness Design in MOP

When multiple objectives $\{f_i(M)\}_{i=1}^k$ must be optimized simultaneously, the task can be formulated as a MOP. While one can form a weighted sum $Fitness_{\text{ws}} = \sum_i \lambda_i f_i$, fixed weights $\lambda_i$ skew search toward certain regions and indirectly affect constraint satisfaction. In contrast, Pareto optimization uncovers the full trade-off front without pre-assigned weights, allowing decision-makers to adjust priorities post hoc.

DEMO uses SPEA2 Zitzler et al. (2001) with CDP Deb et al. (2002) dominance: $M_1 \succ_c M_2$ if $CV(M_1) < CV(M_2)$ or $CV(M_1) = CV(M_2)$ and $M_1 \succ M_2$. Each individual $M_i$ has strength $S_i = |\{j : M_i \succ_c M_j\}|$, raw fitness $RF_i = \sum_{M_j \succ_c M_i} S_j$, density $D_i = 1/(\sigma_i^p + 2)$ where $\sigma_i^p$ is its $p$th-nearest-neighbor distance, and overall fitness $Fitness = RF + D$. where $RF$ ensures convergence and $D$ ensures diversity.

## 4 Experiments

### 4.1 Experimental Setup

**Tasks and Datasets.** We evaluate DEMO on a comprehensive suite of single-/multi-objective molecular design tasks: single-property targeting (SOP-ST), multi-property targeting (SOP-MT), multi-objective optimization (MOP-MO), and constrained multi-objective optimization (MOP-CMO). All experiments leverage the GeoLDM Xu et al. (2023) and EDM Hoogeboom et al. (2022) models pretrained on the QM9 Ramakrishnan et al. (2014) dataset. For the ligand generation task (SOP-CSO), we use models pretrained on GEOM-Drugs Axelrod & Gomez-Bombarelli (2022) and test on 100 protein pockets from CrossDocked2020 Francoeur et al. (2020). To prevent information leakage, the QM9 training set was split equally for predictor training and diffusion model training.

**Metrics.** We report mean absolute error (MAE) for SOP tasks and docking scores from Qvina Hassan et al. (2017) for ligand generation. For MOP tasks, we use Pareto hypervolume (HV) to assess the quality of the trade-off front.

**Baselines.** Our primary principle for baseline selection was to prioritize training-free methods to ensure a fair comparison with our retraining-free approach. We compare DEMO against several classes of baselines: (1) conditional models requiring retraining per task (cEDM, cGeoLDM, GCDM Morehead & Cheng (2024), EEGSDE Bao et al. (2022)); and (2) training-free guidance methods (MUDM Han et al. (2023), TFG Ye et al. (2024)). Both MUDM and TFG are limited as they rely on differentiable evaluators; TFG is further restricted to SOP-ST tasks. Since MUDM is not open-source, we cite its results directly from the original paper. For MOP tasks, where few specialized baselines exist, we compare against strong unconditional generation baselines (EDM, GeoLDM). To ensure fairness, all baseline comparisons are conducted under two budgets: same runtime (SR) and same number of evaluations (SE).

To ensure statistical robustness, all experiments were independently run 20 times. Detailed hyperparameters, hardware specifications, and specifics of the HV calculation are provided in Appendix A.3.

### 4.2 Results on SOP

The experimental results in Table 1 evaluate performance on the single-target inverse design tasks (SOP-ST), where the goal is to discover a 3D structure that precisely matches a given target property value. Our proposed DEMO framework, when combined with various base models, demonstrates exceptional performance by consistently securing the best or second-best results across nearly all evaluated properties, significantly outperforming both standard conditional models (e.g., cEDM) and other training-free guidance methods (e.g., MUDM). Crucially, DEMO exhibits superior search efficiency compared to a generate-and-screen baseline (TopN). While the screening approach remains somewhat competitive when allocated the same number of evaluations (SE), its performance deteriorates sharply under the more practical constraint of the same runtime (SR). In contrast, DEMO maintains its state-of-the-art performance under both budgets, highlighting its efficient evolutionary

Table 1: MAE ($\downarrow$) on SOP-ST. Screening baselines (TopN) are reported as Same Runtime (SR) / Same Number of Evaluations (SE). Best and second-best results are in bold and underlined.

| Method | $\alpha$ | $\Delta\varepsilon$ | $\varepsilon_{homo}$ | $\varepsilon_{lumo}$ | $\mu$ | $C_v$ |
|---|---|---|---|---|---|---|
| Random | 9.01 | 1470 | 645 | 1457 | 1.616 | 6.857 |
| Atoms | 3.86 | 866 | 426 | 813 | 1.053 | 1.971 |
| cEDM | 2.76 | 655 | 356 | 584 | 1.111 | 1.101 |
| cGeoLDM | 2.37 | 587 | 340 | 522 | 1.108 | 1.025 |
| cGCDM | 1.99 | 595 | 346 | 480 | 0.855 | 0.698 |
| EEGSDE | 2.50 | 487 | 302 | 447 | 0.777 | 0.941 |
| TFG | 3.90 | 893 | 984 | 568 | 1.330 | 2.770 |
| MUDM | 0.43 | 85 | 72 | 133 | 0.333 | 0.290 |
| TFG+TopN (SR / SE) | 2.81 / 0.52 | 550 / 132 | 322 / 148 | 407 / 87 | 0.740 / 0.250 | 1.840 / 0.340 |
| EDM+TopN (SR / SE) | 2.43 / 0.41 | 472 / 135 | 318 / 143 | 441 / 89 | 0.630 / 0.210 | 1.690 / 0.300 |
| GeoLDM+TopN (SR / SE) | 3.21 / 1.03 | 536 / 150 | 263 / 60 | 641 / 949 | 0.870 / 0.250 | 3.250 / 0.970 |
| DEMO+GeoLDM | 0.41 | 114 | **23** | 510 | 0.190 | 0.480 |
| DEMO+EDM | **0.12** | 114 | 48 | 57 | **0.080** | 0.190 |
| DEMO+TFG | 0.18 | **58** | 42 | **44** | 0.210 | **0.180** |

Table 2: Comprehensive MAE ($\downarrow$) for the SOP-MT task. Screening baselines (TopN), an ablation study, and DEMO results are reported for two distinct backbones.

| Property Pair | Baselines | | | | Backbone: GeoLDM | | | | | Backbone: EDM | | | | |
|---|---|---|---|---|---|---|---|---|---|---|---|---|---|---|
| | | | | | TopN | | DEMO Ablation | | DEMO | TopN | | DEMO Ablation | | DEMO |
| (Prop1 / Prop2) | MUDM | EEGSDE | EDM | GeoLDM | (SE) | (SR) | w/o CO | w/o MT | (Ours) | (SE) | (SR) | w/o CO | w/o MT | (Ours) |
| $C_v / \mu$ | 1.47/0.69 | 0.98/0.91 | 1.08/1.16 | 1.23/1.12 | 1.75/0.59 | 3.62/1.20 | **0.66**/0.50 | 2.20/0.56 | 1.14/**0.47** | 1.68/0.48 | 3.86/1.17 | **0.66**/0.43 | 2.44/0.59 | 1.23/**0.42** |
| $\Delta\varepsilon / \mu$ | 544/0.58 | 563/0.87 | 683/1.13 | 664/1.13 | 239/0.60 | 494/1.34 | 269/0.60 | 240/0.59 | **188**/**0.42** | 264/0.69 | 546/1.58 | **214**/**0.41** | 238/0.63 | **168**/0.47 |
| $\alpha / \mu$ | 1.32/0.52 | 2.61/0.86 | 2.76/1.16 | 2.77/1.09 | 1.87/0.53 | 4.01/1.17 | **1.00**/0.54 | 2.41/**0.50** | 1.41/0.57 | 1.96/**0.42** | 4.67/1.04 | **0.80**/0.44 | 2.52/0.54 | **1.30**/0.52 |
| $\varepsilon_{homo}/\varepsilon_{lumo}$ | **317**/455 | 355/517 | 372/594 | 384/634 | 353/605 | 489/742 | 372/707 | 494/659 | 319/**407** | 560/896 | 624/1199 | 365/596 | 383/443 | **295**/**410** |
| $\varepsilon_{lumo} / \mu$ | 575/**0.50** | 526/0.86 | 610/1.14 | 636/1.06 | 604/1.56 | 793/1.92 | 583/1.22 | 458/1.15 | **305**/0.87 | **342**/0.96 | 563/1.78 | 681/1.30 | 555/1.23 | 360/0.94 |
| $\varepsilon_{lumo}/\Delta\varepsilon$ | 361/228 | 546/589 | 1097/712 | 457/548 | 567/621 | 724/700 | 627/561 | **294**/346 | 443/406 | 642/548 | 975/650 | 537/635 | 378/382 | 457/452 |
| $\varepsilon_{homo}/\Delta\varepsilon$ | 262/489 | 567/323 | 578/655 | 361/657 | 156/256 | 301/461 | **83**/**172** | 140/188 | 102/176 | 126/201 | 259/464 | **103**/165 | 138/221 | 113/**142** |

search strategy. This demonstrates that DEMO not only discovers molecules that more accurately meet the specified property targets but also achieves this with greater efficiency within a fixed computational budget, underscoring its potential as a flexible and powerful optimization framework.

The results in Table 2 highlight the difficulty of the SOP-MT task, where our DEMO framework consistently emerges as a top performer. Unlike MUDM, which requires pre-defined weights and differentiable objectives, DEMO operates as a true black-box optimizer, a crucial advantage when such prior knowledge is unavailable. In contrast, the TopN screening baseline is not competitive; its time-constrained version (SR-TopN) collapses completely, revealing its inefficiency and that its occasional success is merely a sampling artifact. DEMO's success stems from its role as an efficient navigator of the property landscape, not just a data sampler, making it a more robust discovery tool.

Table 3 evaluates DEMO on protein-ligand generation, focusing on the Vina score. In a standard unconstrained task, DEMO (with GeoLDM/EDM) outperforms the TopN (SE) baseline with scores of -6.39/-6.35 vs. -6.17/-6.15. Furthermore, in a constrained 'Acyclic' task—a strategy to improve synthetic accessibility and flexibility—DEMO again leads with scores of -5.03/-4.94 and achieves an excellent acyclic feasibility of 94.4%/97.8%. This ability to enforce explicit structural rules is a key advantage (details in Appendix A.3.4).

Table 3: Performance on ligands generation.

| Method | Vina-mean ($\downarrow$) | Acyclic% ($\uparrow$) |
|---|---|---|
| TopN+GeoLDM (SE) | -6.17 | – |
| TopN+EDM (SE) | -6.15 | – |
| DEMO+GeoLDM | **-6.39** | – |
| DEMO+EDM | **-6.35** | – |
| TopN (SE)+GeoLDM - Acyclic | -4.80 | 0.731 |
| TopN (SE)+EDM - Acyclic | -4.75 | 0.704 |
| DEMO+GeoLDM - Acyclic | **-5.03** | **0.944** |
| DEMO+EDM - Acyclic | **-4.94** | **0.978** |

## 4.3 RESULTS ON MOP

The Pareto optimization results in Table 4 underscore the superiority of DEMO's active exploration strategy. In the unconstrained (MOP-MO) task, DEMO consistently achieves the highest Hypervolume (HV) in all 15 instances. Its advantage becomes even more stark in the challenging constrained (MOP-CMO) task, where the TopN screening baseline's performance collapses. This highlights a fundamental divide: passive sampling like TopN is limited by the quality and diversity of the

Table 4: HV ($\uparrow$) mean (std) comparison across MOP methods with GeoLDM and EDM backbones. Symbols '+', '−', and '=' indicate the estimated significance of performance relative to the full DEMO method within each backbone section. For visualization results, see A.3.3 and A.3.5

| Property Pair | Backbone: DEMO-GeoLDM | | | | | Backbone: DEMO-EDM | | | | |
|---|---|---|---|---|---|---|---|---|---|---|
| | TopN(SE) | TopN(SR) | w/o CO | w/o MT | DEMO | TopN(SE) | TopN(SR) | w/o CO | w/o MT | DEMO |
| *Unconstrained Multi-Objective Optimization (MOP-MO)* | | | | | | | | | | |
| $\alpha-\Delta\varepsilon$ | $0.617_{(0.033)}^{-}$ | $0.552_{(0.033)}^{-}$ | $0.621_{(0.035)}^{-}$ | $\underline{0.777_{(0.012)}^{=}}$ | $\mathbf{0.778_{(0.017)}}$ | $0.595_{(0.023)}^{-}$ | $0.547_{(0.032)}^{-}$ | $0.618_{(0.037)}^{-}$ | $\mathbf{0.774_{(0.011)}^{=}}$ | $0.771_{(0.017)}$ |
| $\alpha-\varepsilon_{homo}$ | $0.476_{(0.039)}^{-}$ | $0.402_{(0.043)}^{-}$ | $0.479_{(0.048)}^{-}$ | $\underline{0.753_{(0.022)}^{=}}$ | $\mathbf{0.755_{(0.019)}}$ | $0.489_{(0.046)}^{-}$ | $0.421_{(0.060)}^{-}$ | $0.471_{(0.072)}^{-}$ | $0.736_{(0.014)}^{=}$ | $\mathbf{0.740_{(0.003)}}$ |
| $\alpha-\varepsilon_{lumo}$ | $0.405_{(0.027)}^{-}$ | $0.364_{(0.031)}^{-}$ | $0.406_{(0.037)}^{-}$ | $\underline{0.525_{(0.007)}^{-}}$ | $\mathbf{0.528_{(0.006)}}$ | $0.406_{(0.026)}^{-}$ | $0.354_{(0.025)}^{-}$ | $0.392_{(0.041)}^{-}$ | $\mathbf{0.529_{(0.005)}^{=}}$ | $0.528_{(0.006)}$ |
| $\alpha-\mu$ | $0.891_{(0.065)}^{-}$ | $0.809_{(0.065)}^{-}$ | $0.847_{(0.061)}^{-}$ | $\mathbf{1.117_{(0.004)}^{=}}$ | $1.115_{(0.004)}$ | $0.875_{(0.052)}^{-}$ | $0.769_{(0.045)}^{-}$ | $0.857_{(0.066)}^{-}$ | $\mathbf{1.117_{(0.003)}^{=}}$ | $1.110_{(0.012)}$ |
| $\alpha-C_v$ | $0.462_{(0.072)}^{-}$ | $0.357_{(0.085)}^{-}$ | $0.407_{(0.065)}^{-}$ | $\mathbf{0.682_{(0.012)}^{=}}$ | $\underline{0.679_{(0.014)}}$ | $0.414_{(0.054)}^{-}$ | $0.321_{(0.044)}^{-}$ | $0.388_{(0.069)}^{-}$ | $\mathbf{0.675_{(0.006)}^{+}}$ | $\underline{0.664_{(0.014)}}$ |
| $\Delta\varepsilon-\varepsilon_{homo}$ | $0.526_{(0.033)}^{-}$ | $0.473_{(0.045)}^{-}$ | $0.541_{(0.041)}^{-}$ | $0.666_{(0.013)}^{=}$ | $\mathbf{0.667_{(0.017)}}$ | $0.521_{(0.028)}^{-}$ | $0.475_{(0.028)}^{-}$ | $0.534_{(0.030)}^{-}$ | $\mathbf{0.665_{(0.016)}^{=}}$ | $0.663_{(0.016)}$ |
| $\Delta\varepsilon-\varepsilon_{lumo}$ | $0.457_{(0.015)}^{-}$ | $0.431_{(0.012)}^{-}$ | $0.453_{(0.015)}^{-}$ | $0.506_{(0.012)}^{=}$ | $\mathbf{0.509_{(0.012)}}$ | $0.455_{(0.016)}^{-}$ | $0.429_{(0.016)}^{-}$ | $0.468_{(0.019)}^{-}$ | $0.501_{(0.010)}^{-}$ | $\mathbf{0.508_{(0.010)}}$ |
| $\Delta\varepsilon-\mu$ | $\underline{0.982_{(0.016)}^{=}}$ | $0.949_{(0.022)}^{-}$ | $0.971_{(0.022)}^{-}$ | $1.001_{(0.017)}^{-}$ | $\mathbf{1.008_{(0.027)}}$ | $0.980_{(0.012)}^{-}$ | $0.942_{(0.015)}^{-}$ | $0.960_{(0.021)}^{-}$ | $0.998_{(0.019)}^{-}$ | $\mathbf{1.008_{(0.026)}}$ |
| $\Delta\varepsilon-C_v$ | $0.460_{(0.054)}^{-}$ | $0.379_{(0.054)}^{-}$ | $0.489_{(0.067)}^{-}$ | $\mathbf{0.651_{(0.020)}^{=}}$ | $\underline{0.647_{(0.025)}}$ | $0.449_{(0.034)}^{-}$ | $0.397_{(0.048)}^{-}$ | $0.508_{(0.074)}^{-}$ | $0.644_{(0.020)}^{-}$ | $\mathbf{0.656_{(0.019)}}$ |
| $\varepsilon_{homo}-\varepsilon_{lumo}$ | $0.353_{(0.021)}^{-}$ | $0.317_{(0.026)}^{-}$ | $0.374_{(0.022)}^{-}$ | $\mathbf{0.469_{(0.013)}^{=}}$ | $\underline{0.463_{(0.015)}}$ | $0.351_{(0.026)}^{-}$ | $0.320_{(0.020)}^{-}$ | $0.371_{(0.029)}^{-}$ | $\mathbf{0.468_{(0.012)}^{+}}$ | $0.457_{(0.010)}$ |
| $\varepsilon_{homo}-\mu$ | $0.756_{(0.043)}^{-}$ | $0.686_{(0.053)}^{-}$ | $0.745_{(0.050)}^{-}$ | $\underline{0.947_{(0.037)}^{-}}$ | $\mathbf{0.951_{(0.040)}}$ | $0.774_{(0.054)}^{-}$ | $0.668_{(0.035)}^{-}$ | $0.750_{(0.080)}^{-}$ | $0.934_{(0.021)}^{=}$ | $\mathbf{0.935_{(0.246)}}$ |
| $\varepsilon_{homo}-C_v$ | $0.374_{(0.066)}^{-}$ | $0.288_{(0.041)}^{-}$ | $0.384_{(0.089)}^{-}$ | $\underline{0.611_{(0.022)}^{-}}$ | $\mathbf{0.631_{(0.031)}}$ | $0.364_{(0.058)}^{-}$ | $0.302_{(0.054)}^{-}$ | $0.381_{(0.069)}^{-}$ | $\mathbf{0.607_{(0.026)}^{=}}$ | $0.602_{(0.022)}$ |
| $\varepsilon_{lumo}-\mu$ | $0.635_{(0.010)}^{-}$ | $0.611_{(0.016)}^{-}$ | $0.632_{(0.021)}^{-}$ | $\mathbf{0.693_{(0.009)}^{+}}$ | $\underline{0.684_{(0.015)}}$ | $0.636_{(0.011)}^{-}$ | $0.607_{(0.013)}^{-}$ | $0.630_{(0.015)}^{-}$ | $\mathbf{0.693_{(0.010)}^{=}}$ | $0.683_{(0.013)}$ |
| $\varepsilon_{lumo}-C_v$ | $0.330_{(0.046)}^{-}$ | $0.278_{(0.050)}^{-}$ | $0.306_{(0.057)}^{-}$ | $\mathbf{0.437_{(0.013)}^{=}}$ | $\underline{0.439_{(0.020)}}$ | $0.310_{(0.022)}^{-}$ | $0.273_{(0.031)}^{-}$ | $0.340_{(0.052)}^{-}$ | $\mathbf{0.445_{(0.007)}^{=}}$ | $0.441_{(0.017)}$ |
| $\mu-C_v$ | $0.616_{(0.054)}^{-}$ | $0.513_{(0.076)}^{-}$ | $0.652_{(0.085)}^{-}$ | $\mathbf{0.887_{(0.032)}^{+}}$ | $0.879_{(0.034)}$ | $0.619_{(0.065)}^{-}$ | $0.540_{(0.083)}^{-}$ | $0.629_{(0.125)}^{-}$ | $\mathbf{0.892_{(0.021)}^{=}}$ | $0.886_{(0.034)}$ |
| **+/−/=** | 0/14/1 | 0/15/0 | 0/14/1 | 2/1/12 | – | 0/15/0 | 0/15/0 | 0/15/0 | 2/2/11 | – |
| *Constrained Multi-Objective Optimization with Structural Constraints (MOP-CMO), Format: HV / Feasibility Rate* | | | | | | | | | | |
| $\varepsilon_{lumo}-\mu$ | N/A | | $0.352/0.09^{-}$ | $\underline{0.421/0.44^{-}}$ | $\mathbf{0.521/0.93}$ | N/A | | $0.361/0.14^{-}$ | $\underline{0.410/0.49^{-}}$ | $\mathbf{0.533/0.92}$ |
| $\alpha-\Delta\varepsilon$ | N/A | | $0.371/0.08^{-}$ | $\underline{0.420/0.62^{-}}$ | $\mathbf{0.600/0.88}$ | N/A | | $0.351/0.18^{-}$ | $\underline{0.420/0.60^{-}}$ | $\mathbf{0.612/0.91}$ |
| $\mu-C_v$ | N/A | | $0.354/0.08^{-}$ | $\underline{0.384/0.42^{-}}$ | $\mathbf{0.425/0.93}$ | N/A | | $0.388/0.22^{-}$ | $\underline{0.398/0.49^{-}}$ | $\mathbf{0.451/0.91}$ |
| $\varepsilon_{homo}-\varepsilon_{lumo}$ | N/A | | $0.230/0.15^{-}$ | $\underline{0.314/0.44^{-}}$ | $\mathbf{0.320/0.88}$ | N/A | | $0.250/0.18^{-}$ | $\underline{0.308/0.51^{-}}$ | $\mathbf{0.315/0.92}$ |
| **+/−/=** | 0/4/0 | 0/4/0 | 0/4/0 | – | | 0/4/0 | 0/4/0 | 0/4/0 | – | |

pre-generated pool, rendering it incapable of navigating complex constraints or discovering solutions in sparsely populated regions of the property space. DEMO, however, leverages its evolutionary operators to conduct a directed search. This mechanism actively constructs new, high-quality molecules that satisfy strict constraints while simultaneously pushing the population towards the Pareto front—a creative capability entirely absent in passive screening.

Our ablation study reveals that the roles of Crossover (CO) and Mutation (MT) are nuanced and task-dependent (Tables 2 and 4). For the local-search-intensive SOP-MT task, the w/o CO variant excels, highlighting Mutation's critical role in fine-grained refinement. Conversely, for the exploration-focused MOP tasks, the w/o MT variant is often superior, underscoring Crossover's importance as the primary engine for global exploration. Ultimately, DEMO's state-of-the-art performance stems from its synergy: Crossover acts as a global explorer, while Mutation provides local refinement. This balance creates a versatile strategy that excels by adeptly managing the trade-off between exploration and exploitation.

## 5 SUMMARY

In this work, we introduce DEMO, a novel evolutionary framework that overcomes fundamental challenges in 3D molecular optimization. By synergizing the exploratory power of evolutionary algorithms (EAs) with the robust generative capabilities of diffusion models, DEMO effectively navigates complex, multi-objective, and constrained chemical spaces. Our core innovation is a noise-space crossover operator, which decouples the search for optimal properties from the enforcement of chemical validity. We validated DEMO's superiority across a spectrum of challenging tasks, from inverse design (SOP) to constrained Pareto front exploration (MOP-CMO). Our results prove that DEMO can explore property frontiers and adapt to demanding structural constraints, overcoming the limitations of passive, sampling-based methods. Ablation studies confirmed the synergy of its operators, establishing crossover as the engine for global exploration and mutation as the mechanism for precise local refinement. Beyond its function as an optimizer, DEMO serves as a data discovery engine, initiating a virtuous cycle for scientific discovery. By finding novel molecules at the frontiers of chemical space, it generates valuable out-of-distribution data to train more powerful generative models. This self-improving loop elevates DEMO from a static tool to a dynamic engine within a larger, continuously learning discovery system, representing a significant step forward in automated molecular design.

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

# A  APPENDIX

## A.1  DETAILS OF DEMO

### A.1.1  FITNESS EVALUATION AND IMPLICIT VALIDITY CONSTRAINT

In all optimization tasks performed by DEMO, a fundamental, implicit constraint is enforced by default: the requirement that any generated molecule must be chemically valid and structurally feasible. This is not treated as a separate objective but is integrated directly into the fitness evaluation through the constraint violation term, $CV(M)$. A molecule that fails this validity check—for instance, due to incorrect bond lengths, atomic clashes, or other chemically implausible features as determined by a standard validator like RDKit—is assigned a significant penalty.

This penalty is formalized as the validity constraint term, $c_{\text{valid}}(M)$, which is calculated as follows:

$$c_{\text{valid}}(M) = 1 - I_{\text{valid}}(M) \tag{10}$$

where $I_{\text{valid}}(M)$ is an indicator function that returns 1 if the molecule $M$ is deemed chemically valid, and 0 otherwise. This ensures that any invalid molecule receives a non-zero penalty. This term is then included as a component of the total constraint violation score, $CV(M)$, strongly biasing the evolutionary search towards regions of the chemical space that correspond to realistic molecules and effectively pruning invalid candidates from the population during selection.

### A.1.2  CROSSOVER OPERATOR FOR VARIABLE-LENGTH MOLECULES

A key challenge in applying crossover to molecular structures is handling their variable lengths. A naive recombination of two molecules with different numbers of atoms can easily result in a chemically nonsensical offspring. DEMO addresses this with a two-step process to ensure the structural and genetic integrity of the generated children.

1. **Offspring Length Determination:** First, a valid length for the child molecule ($L_{\text{child}}$) is determined. This length is chosen from a permissible range that is defined by the lengths of the two parent molecules ($L_{p1}$ and $L_{p2}$) and the maximum molecular size observed in the pre-training dataset. This step ensures that the resulting offspring will have a size that is both genetically related to its parents and consistent with the general scale of molecules the base model was trained on.

2. **Fragment-based Recombination:** Second, once $L_{\text{child}}$ is set, the offspring's atomic structure is constructed by combining fragments from the two parents. In our implementation, this is analogous to a one-point crossover. A random crossover point is selected, and the initial segment of atoms is taken from the first parent, while the remaining segment is taken from the second parent. The lengths of these two segments are chosen such that their sum equals the predetermined child length, $L_{\text{child}}$. This method guarantees that the offspring has a valid, pre-determined size while still inheriting a substantial and contiguous block of genetic material from both parents, facilitating a meaningful exploration of the chemical space.

### A.1.3  NOISE LEVEL $t'$ SELECTION FOR GEOM-DRUGS DATASET

To determine the optimal noise level $t'$ for operations on the larger and more complex molecules of the GEOM-Drugs dataset, we conducted an empirical study analogous to the one on QM9. We systematically varied $t'$ from 0 to 400 and evaluated the quality of the resulting molecules after mutation and crossover operations using both EDM and GeoLDM as backbones. The results for Atom Stability (AS) and Validity are presented in Figure 5.

Our analysis reveals two key findings:

- **For Mutation:** Similar to QM9, the mutation operation (denoising a noised parent) is robust. Both EDM and GeoLDM maintain high Atom Stability and Validity for $t' \geq 40$. This is because mutation is a local perturbation, and the model can easily recover the original valid structure.

Figure 5: Quality trends for molecules reconstructed from noised parents (Mutation) and noised offspring (Crossover) at varying noise levels $t'$ on the GEOM-Drugs dataset. The metrics show that a higher $t'$' is necessary for crossover to yield high-quality molecules compared to the QM9 dataset.

- **For Crossover:** Recombining two different molecules via crossover is a significantly more disruptive operation, especially for the larger structures in GEOM-Drugs. As shown in the figure, a low $t'$' results in poor-quality offspring. A much higher noise level is required to sufficiently "smooth" the manifold, allowing the denoising model to successfully generate a valid, stable hybrid molecule. The performance of crossover operations begins to saturate at a high level only when $t' \geq 250$'.

Consequently, for experiments involving the GEOM-Drugs dataset, we selected these more conservative thresholds ($t'$=40 for mutation, $t'$=250' for crossover) to ensure the generation of high-quality candidate molecules.

For the GEOM-Drugs analysis, we deliberately omitted the Molecular Stability (MS) metric. The primary reason is that for the larger, more complex molecules typical of this dataset, the combination of the **Atom Stability (AS)** metric and the RDKit-based **Validity** check provides a sufficiently stringent and comprehensive assessment of a molecule's integrity. The AS metric ensures that local atomic environments are chemically correct, while the Validity check enforces global chemical rules (e.g., valency, aromaticity, and connectivity as a single graph). Together, these two metrics serve as a robust proxy for overall molecular stability, making a separate MS metric largely redundant and computationally inefficient for this specific dataset.

### A.1.4 INTEGRATION WITH TRAINING-FREE GUIDANCE (TFG)

Our framework can also be synergistically combined with guidance methods such as Training-Free Guidance (TFG). However, this integration is primarily applicable to Single-Objective Property Targeting (SOP-ST) tasks, as TFG is inherently designed to guide the generation process towards a single, differentiable target property. In this hybrid approach, we apply TFG's gradient-based guidance directly to the noised offspring, $M_{t'}$, immediately after the crossover operation.

Interestingly, our experiments revealed that the choice of noise level $t'$ is critical for the success of this integration. We found that naively using the same fixed noise level as our unguided experiments ($t' = 200$) resulted in a degradation of performance (achieving a MAE of 0.44 for the $\mu$ property and similarly poor results for others: $126, 81, 0.12, 0.28$). In contrast, the superior results reported in the main paper for the DEMO+TFG variant (MAE of 0.18, and values of $58, 42, 44, 0.210, 0.180$) were achieved using a dynamic noise schedule with $t'_{\max} = 400$ and $t'_{\min} = 200$.

We surmise that the reason for this behavior lies in the state of the population during different evolutionary stages. In the early generations, the population is highly diverse and molecules are likely to be far from the target property region. Applying a low noise level like $t' = 200$ adds only a small perturbation, meaning the potential manifold of clean molecules ($M_0$) that can be recovered from $M_{t'}$ is highly constrained and localized. In such cases, TFG's gradient may be insufficient to guide the denoising process to the distant target region. This can even be detrimental, as an incomplete guidance process might terminate prematurely, yielding an offspring that has been distorted from a valid structure but has not yet reached the target property.

Therefore, a dynamic schedule is more effective. In the early, exploratory stages, a larger $t'$ (e.g., 400) adds more substantial noise, expanding the potential recovery space and providing the TFG gradient with a wider manifold to successfully navigate towards the target. Conversely, in later generations, when the population has already converged near the target region, a smaller $t'$ (e.g., 200) is preferable. It preserves more of the high-quality structural information from the parent molecules while still allowing for fine-grained adjustments guided by TFG to precisely hit the target value.

### A.1.5 ENFORCING COMPLEX STRUCTURAL CONSTRAINTS

In this section, we describe our methodology for optimizing molecules subject to the inclusion of a specific, and potentially complex, structural fragment. Our goal is to emulate real-world **materials discovery scenarios, from drug design to the engineering of quantum materials**. These applications often require preserving not only a key functional core (such as a chromophore in an organic semiconductor or a specific coordination complex) but also larger structural motifs that are critical for dictating overall properties like electronic band structure, thermal stability, or crystal packing. To model this complexity, our target structures are **randomly selected connected fragments** from existing molecules. This makes the matching task more challenging and practical than simply searching for common, simple functional groups, as desired motifs in materials science are often large and non-canonical.

A key challenge in this process is that establishing a clear Structure-Property Relationship (SPR) is notoriously difficult. The relationship between a specific structural fragment and multiple, often conflicting, target properties is typically a highly complex, non-linear, and non-differentiable function. Modifying one part of a molecule to improve one property can have unforeseen and detrimental effects on others. While deep learning models can be trained to approximate this function, they require vast amounts of labeled data and must be retrained for any new property or structural constraint.

Evolutionary Algorithms (EAs) are ideally suited to circumvent this difficulty. As black-box optimizers, they do not need an explicit model of the SPR; they only require a fitness function to evaluate candidate solutions. This allows them to effectively navigate complex fitness landscapes by treating the structure-property mapping as an unknown function to be optimized. A major advantage of our DEMO framework is its ability to leverage this EA principle **without any retraining**. Instead of explicitly modeling the SPR, DEMO converts this challenge into a multi-objective fitness landscape where the evolutionary algorithm automatically balances the trade-offs.

However, directly applying traditional EAs to 3D molecular structures is itself a major challenge. Standard genetic operators like crossover and mutation, when applied naively to 3D atomic coordinates, almost invariably violate fundamental chemical laws, producing geometrically and chemically invalid offspring. This leads to a highly inefficient search process, as the vast majority of generated candidates must be discarded. **This critical challenge—how to harness the black-box optimization power of EAs in the 3D molecular domain without being crippled by the validity problem—serves as a primary motivation for our DEMO framework.**

To quantify the structural constraint, we developed a two-stage hybrid matching approach that balances computational efficiency with 3D structural accuracy.

1. **Stage 1: 2D Topological Pre-filtering.** A direct, exhaustive 3D subgraph search is computationally prohibitive. Therefore, our first stage serves as a rapid pre-filter. This requires converting the generated 3D molecule into an RDKit 'Chem.Mol' object, which represents a 2D molecular graph. We use RDKit's 'FindMCS()' on this 2D representation to get a topological match ratio, $R_{2D}$. This stage acts only as a filter: a candidate molecule only proceeds to the next stage if $R_{2D}$ exceeds a threshold of 0.8.

2. **Stage 2: Rigorous 3D Geometric Verification.** For candidates that pass the initial check, this stage employs the VF2 algorithm to perform a precise subgraph isomorphism test that accounts for 3D spatial relationships. This step validates the geometric integrity of the fragment, yielding a final 3D match ratio, $R_{3D}$.

The final constraint violation for the structure, $c_{\text{struct}}(M)$, is formulated to penalize imperfections in both the 2D and 3D matching stages. The primary penalty comes from the 3D verification, but

a "soft penalty" is also applied for an imperfect 2D topological match to further guide the search. This is formalized as follows:

$$c_{\text{struct}}(M) = \begin{cases} 1 - R_{\text{2D}} \times 0.5 & \text{if } R_{\text{2D}} < 0.8 \\ 1 - R_{\text{3D}} & \text{if } R_{\text{2D}} \geq 0.8 \end{cases} \tag{11}$$

where $R_{\text{2D}}$ and $R_{\text{3D}}$ are the match ratios from the 'FindMCS' and VF2 algorithms, respectively. This value is then included in the total constraint violation term, $CV(M)$, allowing the evolutionary process to seamlessly co-optimize for structural integrity and other target properties.

### A.1.6 ALGORITHMIC COMPLEXITY ANALYSIS

The computational complexity of the DEMO framework is analyzed on a per-generation basis. For a population of size $N$, the total complexity of one generation can be broken down into the costs of its primary components: evolutionary operations, the reverse diffusion process, and fitness evaluation.

Let $G$ be the total number of generations. The complexity for a single generation is given by: $O(\text{Selection} + \text{Crossover/Mutation} + \text{Denoising} + \text{Fitness} + \text{Environmental Selection})$

The complexity of each component is as follows:

- **Evolutionary Operators:** Parent selection (e.g., binary tournament) and crossover/mutation operations are computationally efficient, with a complexity of $O(N)$. The bottleneck in the evolutionary part is typically the environmental selection, especially for multi-objective optimization (MOP) tasks. Using a standard dominance-based sorting algorithm like SPEA2 or NSGA-II, this step involves pairwise comparisons among the combined population of parents and offspring ($2N$), leading to a complexity of $O((2N)^2) = O(N^2)$.

- **Reverse Diffusion Process (Denoising):** This is one of the main computational drivers of the algorithm. For each of the $N$ candidate molecules in the offspring pool, the model performs a reverse diffusion process starting from a noise level $t'$. This involves $t'$ sequential forward passes through the denoising network. If we denote the complexity of a single forward pass of the diffusion model as $C_{\text{model}}$, the total complexity for this step is $O(N \cdot t' \cdot C_{\text{model}})$.

- **Fitness Evaluation:** The cost of this step is highly dependent on the external property predictors or simulators used. If the complexity of evaluating a single molecule is $C_{\text{fitness}}$, then for the combined population of size $2N$, the total cost is $O(N \cdot C_{\text{fitness}})$. In scenarios involving computationally intensive evaluations like molecular docking, this term can become a dominant factor.

Combining these terms, the overall complexity for a single generation of DEMO is:

$$O(N^2 + N \cdot C_{\text{fitness}} + N \cdot t' \cdot C_{\text{model}})$$

The total complexity for a full run of the algorithm over $G$ generations is therefore:

$$O(G \cdot (N^2 + N \cdot C_{\text{fitness}} + N \cdot t' \cdot C_{\text{model}}))$$

In practical applications, although the Pareto-based environmental selection has a theoretical complexity of $O(N^2)$, its runtime contribution is negligible for the typical, relatively small population sizes used. The primary computational burdens are the fitness evaluation ($N \cdot C_{\text{fitness}}$) and the iterative denoising process ($N \cdot t' \cdot C_{\text{model}}$).

Actual runtimes can be found in A.3.3.

### A.2 THEORETICAL FOUNDATION OF DIRECTIONAL EVOLUTION

The central mechanism enabling DEMO's directional search is not merely heuristic but is grounded in the probabilistic and geometric properties of the diffusion model's latent space. This section provides a theoretical justification for how crossover and mutation operators steer the generative process by manipulating latent distributions and leveraging the denoising network as a learned projection operator.

### A.2.1 MATHEMATICAL FORMULATION OF EVOLUTIONARY OPERATORS

**Probabilistic Interpretation of Crossover.** Let us consider two parent molecules, $M_0^{(1)}$ and $M_0^{(2)}$, selected for their high fitness. The crossover operation produces an offspring latent representation $M_{t'}^{(o)}$ from a new Gaussian distribution whose mean is a scaled, fragment-wise interpolation of the original clean parent molecules:

$$M_{t'}^{(o)} \sim \mathcal{N}\left(\sqrt{\bar{\alpha}_{t'}}\left(B M_0^{(1)} + (1-B)M_0^{(2)}\right), (1-\bar{\alpha}_{t'})I\right) \tag{12}$$

This operation represents a targeted leap into a region of the chemical space biased to inherit successful structural motifs from both parents, making it a powerful tool for global exploration.

**Probabilistic Interpretation of Mutation.** The mutation operation is simpler, involving only a single parent, $M_0^{(1)}$. It is equivalent to adding noise and then denoising, which can be described as sampling from the parent's forward process distribution:

$$M_{t'}^{(\text{mut})} \sim \mathcal{N}\left(\sqrt{\bar{\alpha}_{t'}} M_0^{(1)}, (1-\bar{\alpha}_{t'})I\right) \tag{13}$$

Unlike crossover, the mean of this distribution is a scaled version of a single, structurally valid parent. This frames mutation as a highly localized search in the immediate vicinity of a known high-fitness individual, making it ideal for local refinement.

### A.2.2 GEOMETRIC INTERPRETATION AS MANIFOLD PROJECTION

**The Off-Manifold Nature of the Crossover Mean.** The "chimeric mean" of the crossover distribution (from Eq. 12) is a synthetic point highly likely to be "off-manifold," corresponding to a chemically nonsensical structure. The key to generating a valid molecule is the noise variance, which provides a projection or blurring radius ($\sigma_{t'} = \sqrt{1 - \bar{\alpha}_{t'}}$). When $t'$ is sufficiently large, this radius is also large, ensuring that the sampling hypersphere around the off-manifold mean intersects with the learned data manifold $\mathcal{M}$. The denoiser can then act as a projection operator, mapping a sample from this overlapping region back onto the manifold.

**The On-Manifold Nature of the Mutation Mean.** In stark contrast, the mean of the mutation distribution, $\mu_{t'}^{(\text{mut})} = \sqrt{\bar{\alpha}_{t'}} M_0^{(1)}$, is fundamentally different. Since the parent $M_0^{(1)}$ is a valid molecule and thus lies on the manifold $\mathcal{M}$, its scaled version $\mu_{t'}^{(\text{mut})}$ lies on a scaled version of the same manifold. It is inherently "on-manifold" or, at the very least, in a structurally coherent region of the latent space. Consequently, the sampling hypersphere for mutation is always centered in a "valid" region. This guarantees that its volume will always robustly intersect with the data manifold, regardless of the size of the projection radius.

### A.2.3 THE OPERATOR-SPECIFIC TRADE-OFFS OF $t'$

The selection of $t'$ involves a critical trade-off between the generative flexibility afforded by the projection radius and the information retention from the parents, scaled by $\sqrt{\bar{\alpha}_{t'}}$. However, this trade-off manifests differently for crossover and mutation.

- **Large $t'$**: As $t' \to T$, $\sqrt{\bar{\alpha}_{t'}} \to 0$. The parental signal vanishes for both operators, and the process resembles unconditional generation. Evolutionary guidance is lost.
- **Small $t'$**: As $t' \to 0$, $\sqrt{\bar{\alpha}_{t'}} \to 1$. The parental signal is maximal.

**Implications for Crossover and Mutation.** The geometric nature of the means explains their different requirements for $t'$.

- **For Crossover**, an intermediate $t'$ is a necessity. It must be large enough to provide the geometric flexibility (projection radius) to overcome its off-manifold mean and ensure validity, but small enough so that the parental signal remains strong enough to guide the search.

- **For Mutation**, because its mean is already on-manifold, the risk of generative failure at small $t'$ is virtually eliminated. A small projection radius is sufficient because the center is already in a valid location. Therefore, mutation can operate effectively with a much smaller $t'$. This allows for fine-grained local exploitation and refinement of a promising molecule without losing its high-quality structural information to excessive noise.

This balanced, operator-specific approach to setting $t'$ is what allows DEMO to synergistically combine global, feature-recombining exploration (crossover) with precise, structure-preserving refinement (mutation).

### A.2.4 PROBABILISTIC FRAMEWORK FOR CONVERGENCE

The convergence of DEMO can be understood through a formal probabilistic framework that describes how evolutionary operators guide the search toward high-fitness molecular distributions. For this analysis, let us introduce some clear definitions.

Let $\mathbb{M}$ denote the manifold of all chemically valid molecules. The pretrained diffusion model has learned the data distribution $P_{\text{data}}(M)$ over this manifold. Our optimization goal is to discover molecules within a specific target subset of this manifold, $\mathbb{M}_{\text{target}} \subset \mathbb{M}$, characterized by a high-fitness distribution we denote as $P_{\text{target}}(M)$. By design, the support of $P_{\text{target}}$ is contained within the support of $P_{\text{data}}$. The parent population at any generation can be seen as a set of samples approximating $P_{\text{target}}$.

**Convergence via Crossover.** Consider two high-fitness parents, $M_0^{(1)}, M_0^{(2)} \sim P_{\text{target}}(M)$. The crossover operation generates a noisy latent offspring $M_{t'}^{(o)}$ by sampling from the distribution defined in Eq. 12:

$$P_{t'}^{(o)}(M_{t'}) = \mathcal{N}\left(M_{t'}; \mu^{(o)}, (1 - \bar{\alpha}_{t'})I\right), \quad \text{where} \quad \mu^{(o)} = \sqrt{\bar{\alpha}_{t'}}\left(BM_0^{(1)} + (1 - B)M_0^{(2)}\right). \tag{14}$$

Let us denote the full, deterministic denoising process from time $t'$ to 0 as a function $D_\theta : M_{t'} \mapsto M_0$, where $\theta$ represents the parameters of the denoising network. The evolutionary search makes progress if the denoised offspring, $M_0^{(o)} = D_\theta(M_{t'}^{(o)})$, has a non-zero probability of belonging to the target region $\mathbb{M}_{\text{target}}$.

The challenge arises because the chimeric mean $\mu^{(o)}$ is generally "off-manifold" and thus not a point within $\mathbb{M}_{\text{target}}$. Let $\mathcal{R}_\theta \subset \mathbb{R}^{3 \times n}$ be the region of the latent space from which the denoiser $D_\theta$ can successfully project samples back to the valid manifold $\mathbb{M}$. Crossover is productive only if the support of its sampling distribution $P_{t'}^{(o)}$ has a non-trivial intersection with this "recoverable" region, i.e., $\text{supp}(P_{t'}^{(o)}) \cap \mathcal{R}_\theta \neq \emptyset$.

The noise level $t'$ critically controls this intersection:

- A **large** $t'$ ($\bar{\alpha}_{t'} \to 0$) makes the variance $(1 - \bar{\alpha}_{t'})$ large. The distribution $P_{t'}^{(o)}$ becomes diffuse, increasing the probability of intersecting with $\mathcal{R}_\theta$ and yielding a valid molecule. However, the parental signal, scaled by $\sqrt{\bar{\alpha}_{t'}}$, is attenuated, so the resulting molecule might lack high fitness.

- A **small** $t'$ ($\bar{\alpha}_{t'} \to 1$) concentrates $P_{t'}^{(o)}$ tightly around the off-manifold mean $\mu^{(o)}$. This risks that the entire distribution falls outside of $\mathcal{R}_\theta$, leading to generative failure.

Thus, an intermediate $t'$ is essential to balance the generative reach (ensured by a sufficiently large variance) with the fidelity of parental trait inheritance (encoded in the mean). The mean $\mu^{(o)}$ is the core of this mechanism; it acts as a **directional bias**, anchoring the search in a latent region constructed by interpolating between two known high-fitness points.

This transforms the generative process from a blind, unconditional search into an **informed search**. To see this, let $P_{\text{uncond}}$ be the probability that a sample from the unconditional prior $\mathcal{N}(0, I)$ denoises to a molecule in the desired high-fitness region $\mathbb{M}_{\text{target}}^+$. The core thesis of crossover is that the probability of success for an offspring, $P_{\text{crossover}}$, is substantially higher:

$$P_{\text{crossover}} \gg P_{\text{uncond}} \tag{15}$$

This enhanced probability, which drives the convergence, is formally expressed as:

$$P_{\text{crossover}} = \int_{\mathbb{R}^{3 \times n}} \mathbb{I}[D_\theta(M_{t'}) \in \mathbb{M}_{\text{target}}^+] \cdot P_{t'}^{(o)}(M_{t'}) \, dM_{t'} \tag{16}$$

where $\mathbb{M}_{\text{target}}^+$ is the subset of molecules with fitness exceeding that of the parents. By anchoring the sampling distribution $P_{t'}^{(o)}$ near a promising region, crossover makes the discovery of superior solutions not just possible, but statistically probable.

**Convergence via Mutation.** The mutation process, sampling from the distribution in Eq. 13, serves a complementary role as a principled local exploitation operator. Its mean, $\mu^{(\text{mut})} = \sqrt{\bar{\alpha}_{t'}} M_0^{(1)}$, is inherently "on-manifold," guaranteeing a robust intersection with the recoverable region $\mathcal{R}_\theta$ even for small values of $t'$, where parental information retention is maximal.

The efficacy of mutation as an optimizer is contingent on the local structure of the fitness landscape. Let us formalize this with a key assumption.

**Assumption 1 (Local Fitness Landscape Continuity).** *Let $d(\cdot, \cdot)$ be a distance metric (e.g., RMSD) on the molecular manifold $\mathbb{M}$. For a high-fitness molecule $M_0 \in \mathbb{M}_{\text{target}}$, we assume the fitness function is locally continuous. That is, for a small neighborhood radius $\delta > 0$, there exists a non-trivial subset of molecules $M_0' \in \{M \in \mathbb{M} \mid d(M_0, M) < \delta\}$ for which Fitness($M_0'$) > Fitness($M_0$).*

Mutation operationalizes this assumption. The noise level $t'$ controls the effective search radius $\delta$. A small $t'$ ensures that the generated offspring $M_0^{(\text{mut})} = D_\theta(M_{t'}^{(\text{mut})})$ is structurally close to its parent $M_0^{(1)}$ with high probability. We can state that the expected distance is a monotonically increasing function of $t'$:

$$\mathbb{E}\left[d(M_0^{(1)}, M_0^{(\text{mut})})\right] = f(t'), \quad \text{where} \quad \frac{\partial f}{\partial t'} > 0. \tag{17}$$

By choosing a small $t'$, we constrain the search to a small $\delta$-neighborhood around the parent.

Let us define the *local improvement set* for a parent $M_0^{(1)}$ as:

$$\mathcal{I}(M_0^{(1)}, \delta) = \{M \in \mathbb{M} \mid d(M_0^{(1)}, M) < \delta \text{ and Fitness}(M) > \text{Fitness}(M_0^{(1)})\}. \tag{18}$$

Under Assumption 1, this set is non-empty. The goal of mutation is to generate an offspring that falls within this set. The probability of such a successful local improvement, $P_{\text{mut-improve}}$, is given by the integral of the offspring's effective probability distribution, $P_{\text{eff}}(M_0|M_0^{(1)}, t')$, over this set:

$$P_{\text{mut-improve}} = \int_{\mathcal{I}(M_0^{(1)}, \delta)} P_{\text{eff}}(M_0|M_0^{(1)}, t') \, dM_0 > 0. \tag{19}$$

While this probability is not guaranteed to be large, it is non-zero and, crucially, it is being evaluated in a region already known to have high fitness.

**The Requirement for a Minimal Noise Level.** A subtle but critical point is that the on-manifold nature of the mean does not guarantee generative success for arbitrarily small $t'$. As empirically shown (e.g., in Figs. 4 and 5), validity drops significantly if $t'$ is below a certain threshold. This phenomenon arises because the denoiser $D_\theta$ not a perfect analytical projector but a learned function whose reliability is dependent on the Signal-to-Noise Ratio (SNR) of its input. At extremely small $t'$ (high SNR), the input $M_{t'}^{(\text{mut})}$ is nearly identical to the clean parent. The network is tasked with predicting a near-zero noise vector, a regime where it is often less stable. Minor prediction inaccuracies by $D_\theta$, while small in absolute terms, can be sufficient to violate precise chemical constraints (e.g., bond lengths or angles), pushing the final output off the manifold $\mathbb{M}$. Therefore, mutation requires a minimal noise level, which we denote $\tau_{\min}$ (e.g., $\tau_{\min} \approx 20$ for the QM9 dataset), to operate reliably. This threshold ensures that the input signal is placed within the denoiser's robust operational regime, where it functions effectively as a structural projector rather than an unstable near-identity map. This practical constraint refines our theoretical model: the effective search radius $\delta$ is not just controlled by $t'$, but is bounded from below, ensuring that the local exploitation is both meaningful and, crucially, valid.

This makes mutation a highly efficient, gradient-free method for hill-climbing on the fitness manifold, providing the necessary pressure for the population to converge towards the precise peaks of the target distribution $P_{\text{target}}(M)$.

In summary, DEMO's convergence is driven by the powerful synergy of a biased global search (crossover) and a robust local exploitation mechanism (mutation). Crossover makes informed leaps into promising, interpolated regions, while mutation enables the meticulous exploration of the vicinity of the best-found solutions, facilitating a steady and reliable progression towards optimality.

## A.3 EXPERIMENTAL DETAILS

### A.3.1 IMPLEMENTATION DETAILS

**Hardware.** All experiments were conducted on a single NVIDIA RTX 3090 GPU.

**Baseline Limitations.** We note that the original TFG implementation uses EDM as its backbone, while MUDM uses GeoLDM. Their applicability is limited: TFG is designed exclusively for SOP-ST tasks, and MUDM supports both SOP-ST and SOP-MT. Furthermore, both methods are contingent on the availability of differentiable property evaluators.

**Hyperparameters.** For all evolutionary tasks, we use a population size of $N = 32$. The number of generations was tailored to the task complexity: 10 for SOP-ST, 20 for SOP-MT and MOP-MO, and 25 for the more complex MOP-CMO and ligand generation tasks. The noise schedule for unconditional sampling ($\mathcal{M} = \emptyset$) was fixed at $t_{\min} = t_{\max} = 200$. For TFG, we used a dynamic schedule of $t_{\min} = 200$ and $t_{\max} = 400$. This range was chosen because at $t = 200$, the noisy state $M_{t'}$ is already very similar to the original molecule $M_0$, leaving minimal scope for TFG to provide further improvement, while the higher noise level at the start allows for broader exploration.

### A.3.2 METRIC CALCULATION

**HV Normalization.** To ensure a fair comparison for HV calculations, we first establish the upper and lower bounds for the target objectives using the entire training dataset. These fixed bounds are then used to normalize the objective values for all solutions generated by every method. The HV is subsequently computed on these normalized Pareto fronts, ensuring that the metric reflects the quality of the solutions relative to the known data distribution.

### A.3.3 THE IMPACT OF DIFFERENT POPULATION SIZES

Table 5: Comparison of runtimes (in seconds) between our proposed method and the Top-N (SE) baseline for different population sizes.

| Population Size | Our Method's Runtime (s) | Top-N (SE) Runtime (s) |
|---|---|---|
| 4 | 62.02 | 209.20 |
| 8 | 64.13 | 212.72 |
| 16 | 98.94 | 340.71 |
| 32 | 175.47 | 619.45 |
| 64 | 319.68 | 1116.00 |
| 128 | 620.13 | 2144.37 |

To systematically evaluate the impact of population size and highlight the efficiency of our proposed method, we conducted a series of experiments assessing both performance (Hypervolume, HV) and computational cost (runtime). We tested various population sizes ranging from 4 to 128 (Table 6) and compared our method against a standard Top-N (Same Evaluations, SE) screening baseline.

The experimental results reveal a clear advantage for our approach:

1. **On Performance:** As previously established, larger population sizes consistently achieved the highest HV scores across most tasks. This demonstrates that greater population diversity is crucial for thoroughly exploring the solution space and discovering superior sets of solutions.

2. **On Cost and Efficiency:** As detailed in Table 5, our method demonstrates a significant computational advantage over the Top-N (SE) screening baseline at *every* population size.

The runtime disparity grows substantially with scale. For instance, with a population of 32, our method completes in 175.47 seconds, whereas the Top-N (SE) approach requires 619.45 seconds—over 3.5 times longer.

To further contextualize these findings and evaluate DEMO's exploratory capabilities, we established an additional baseline by first generating a large sample pool (twice the size of the training set) using an unconditional GeoLDM model. These molecules were then evaluated by predictors trained on the other half of the dataset, with the valid results represented as grey dots in our plots. This comparison highlights that while unconditional sampling can generate diverse molecules, its exploration is largely confined to the dense regions of the learned distribution. In contrast, DEMO not only effectively discovers the Pareto Front (PF) but also identifies high-performing data points in regions that are sparsely populated or entirely missed by the unconditional sampling process (Figure 678). This suggests that the base generative model inherently possesses the capability to produce these exceptional solutions, but this potential remains almost completely untapped through standard sampling. **The guidance from DEMO is therefore crucial for unlocking this latent potential and enabling the model to manifest its full generative capabilities.**

This dual advantage—superior search efficiency compared to Top-N screening and enhanced exploratory power beyond unconditional generation—reinforces our conclusion that a population size of 32 is an optimal trade-off point for our experiments. It not only achieves a high HV score with a substantially lower runtime, but its evolutionary approach also intelligently guides the search toward promising and novel regions of the chemical space, in stark contrast to the brute-force 'generate-and-screen' nature of baseline methods.

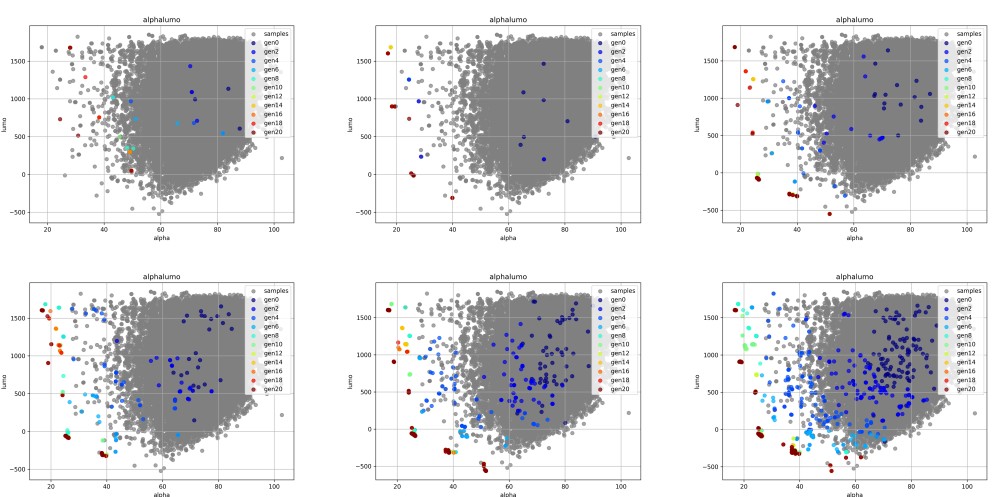

Figure 6: Convergence for different population sizes in the alpha-lumo optimization task. (Gray points are valid solutions among the 100,000 solutions generated using unconditional GeoLDM.)

### A.3.4 METHODOLOGY FOR PROTEIN-LIGAND GENERATION

In this subsection, we detail the methodology for our protein-ligand generation task. First, we employ GEOLDM, pre-trained on the GEOM-Drugs dataset, to generate novel three-dimensional molecular conformations. Subsequently, we utilize Qvina 2.1 to perform docking simulations and evaluate the binding affinity of these conformations. To ensure the chemical and geometric plausibility of the docked poses, all successfully docked molecules are then validated using PoseBuster, configured with the 'dock' checklist as specified in 7.

This validation is not treated as a simple binary pass/fail but is quantified as a penalty. The constraint violation score for the PoseBuster check, $c_{\text{PBvalid}}(M)$, is calculated based on the number of failed criteria from the checklist, as follows:

$$c_{\text{PBvalid}}(M) = |\text{PBChecklist}| - |\text{PBChecklist}_{\text{satisfied}}| \qquad (20)$$

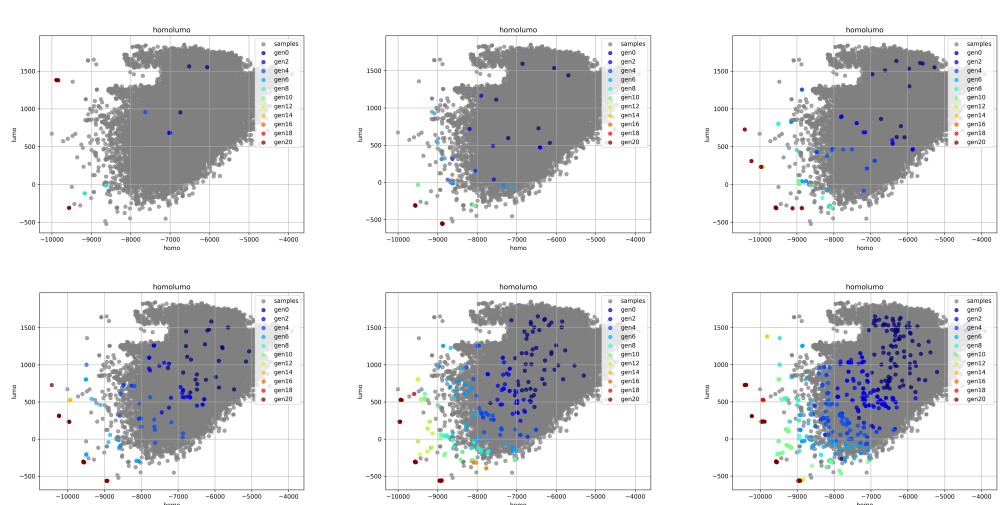

Figure 7: Convergence for different population sizes in the homo-lumo optimization task. (Gray points are valid solutions among the 100,000 solutions generated using unconditional GeoLDM.)

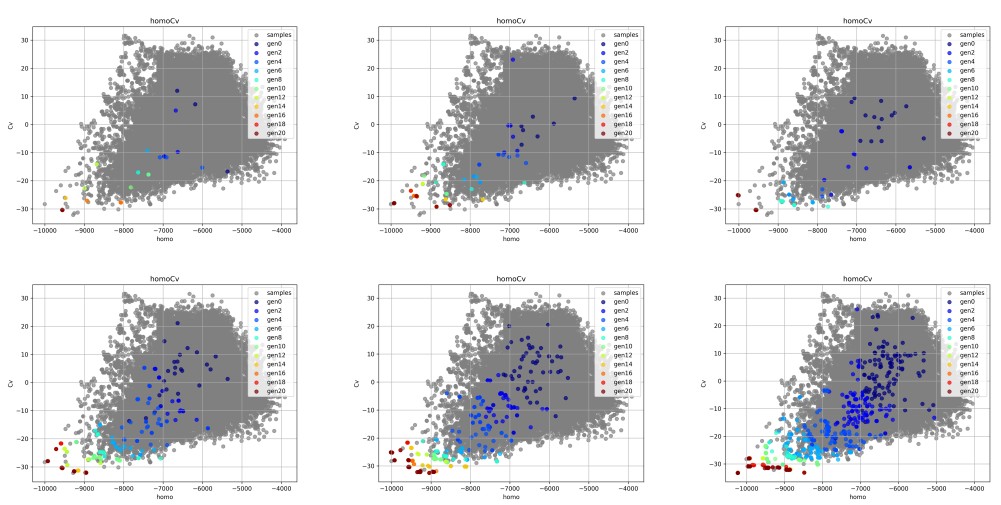

Figure 8: Convergence for different population sizes in the homo-Cv optimization task. (Gray points are valid solutions among the 100,000 solutions generated using unconditional GeoLDM.)

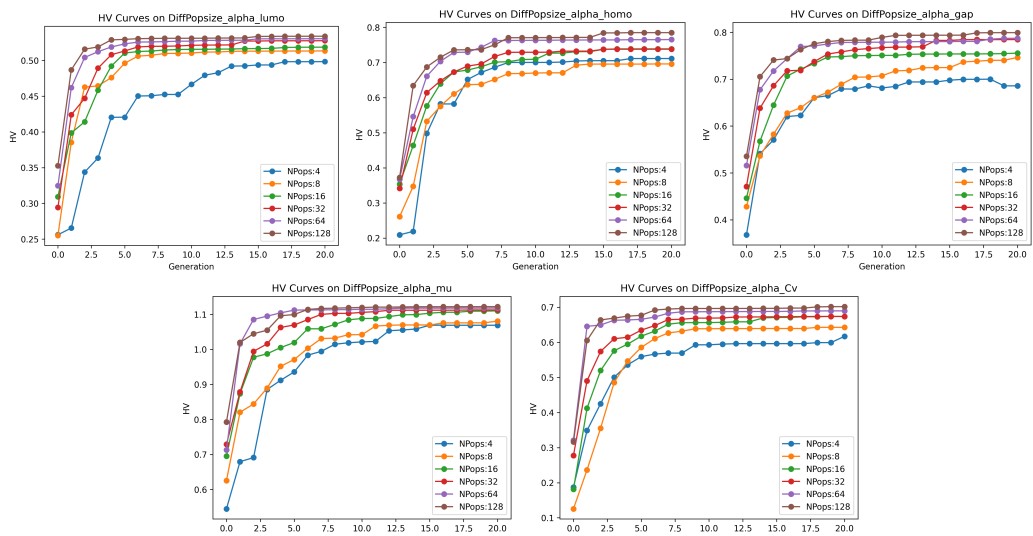

Figure 9: The HV Convergence for different population sizes in multi-objective optimization (Ge-oLDM).

where |PBChecklist| represents the total number of validation checks defined in the checklist, and |PBChecklist$_{\text{satisfied}}$| is the number of those checks that the molecule's docked pose successfully passes. This score, which corresponds to the count of failed checks, is then included as a component of the total constraint violation score, $CV(M)$, thereby penalizing geometrically and chemically implausible binding poses.

In addition to these standard procedures, we introduce a specific task focused on generating acyclic ligands. This focus is motivated by several key physicochemical advantages relevant to drug design. Acyclic ligands typically possess a more flexible molecular backbone compared to their rigid cyclic counterparts. This inherent flexibility is critical, as it allows them to adopt a wider range of conformations, enabling a better **induced fit** within the protein's binding pocket and optimizing interactions with active site residues. From a medicinal chemistry perspective, acyclic compounds are often more **synthetically accessible** and allow for more straightforward functionalization, facilitating the rapid exploration of Structure-Activity Relationships (SAR). Furthermore, the absence of rigid ring systems can **reduce steric hindrance**, potentially improving the ligand's access to and affinity for the binding site.

To enforce this acyclic constraint, we utilize the RDKit library to determine the number of rings in each generated ligand. This integer count is then directly incorporated into the fitness function as a penalty term. The constraint violation, $c_{\text{acyclic}}(M)$, is formalized as:

$$c_{\text{acyclic}}(M) = \text{NumRings}(M) \tag{21}$$

where $\text{NumRings}(M)$ is the function that returns the number of rings in molecule $M$. This value is then included as a component of the total constraint violation score, $CV(M)$, ensuring that any molecule containing one or more rings is penalized proportionally to its degree of cyclicity, thereby guiding the search towards acyclic structures.

### A.3.5 VISUALIZATION RESULTS OF MOP-CMO

### A.4 USE OF LARGE LANGUAGE MODELS

During the preparation of this manuscript, we utilized Large Language Models (LLMs) to assist in polishing the language and improving the clarity of our writing. The core scientific contributions, including the methodology, experimental design, and analysis of results, were conceived and executed entirely by the authors. LLMs were employed solely as a writing aid to enhance readability and ensure grammatical correctness, and did not contribute to the original ideas or technical content of the paper.

Table 6: Mean and standard deviation value of HV($\uparrow$) comparison of different molecular property combinations in different size of Population

| Property 1 | Property 2 | Size of Populaitons | | | | | |
|---|---|---|---|---|---|---|---|
| | | 4 | 8 | 16 | 32 | 64 | 128 |
| $\alpha$ | $\Delta\varepsilon$ | 0.686 (0.018) | 0.746 (0.009) | 0.756 (0.018) | 0.778 (0.017) | 0.787 (0.006) | **0.799** (0.008) |
| $\alpha$ | $\varepsilon_{homo}$ | 0.711 (0.051) | 0.696 (0.034) | 0.739 (0.005) | 0.755 (0.019) | 0.765 (0.028) | **0.785** (0.001) |
| $\alpha$ | $\varepsilon_{lumo}$ | 0.498 (0.007) | 0.513 (0.002) | 0.519 (0.004) | 0.528 (0.006) | 0.530 (0.003) | **0.534** (0.000) |
| $\alpha$ | $\mu$ | 1.046 (0.014) | 1.069 (0.009) | 1.092 (0.006) | 1.115 (0.004) | 1.138 (0.003) | **1.161** (0.002) |
| $\alpha$ | $C_v$ | 0.637 (0.049) | 0.651 (0.032) | 0.665 (0.021) | 0.679 (0.014) | 0.693 (0.009) | **0.707** (0.006) |
| $\Delta\varepsilon$ | $\varepsilon_{homo}$ | 0.626 (0.059) | 0.639 (0.039) | 0.653 (0.026) | 0.667 (0.017) | 0.681 (0.011) | **0.695** (0.007) |
| $\Delta\varepsilon$ | $\varepsilon_{lumo}$ | 0.477 (0.042) | 0.488 (0.028) | 0.498 (0.018) | 0.509 (0.012) | 0.520 (0.008) | **0.530** (0.005) |
| $\Delta\varepsilon$ | $\mu$ | 0.945 (0.095) | 0.966 (0.062) | 0.987 (0.041) | 1.008 (0.027) | 1.029 (0.018) | **1.050** (0.012) |
| $\Delta\varepsilon$ | $C_v$ | 0.607 (0.088) | 0.620 (0.058) | 0.634 (0.038) | 0.647 (0.025) | 0.660 (0.017) | **0.674** (0.011) |
| $\varepsilon_{homo}$ | $\varepsilon_{lumo}$ | 0.434 (0.053) | 0.444 (0.035) | 0.453 (0.023) | 0.463 (0.015) | 0.473 (0.010) | **0.482** (0.006) |
| $\varepsilon_{homo}$ | $\mu$ | 0.892 (0.140) | 0.912 (0.092) | 0.931 (0.061) | 0.951 (0.040) | 0.971 (0.026) | **0.990** (0.017) |
| $\varepsilon_{homo}$ | $C_v$ | 0.592 (0.109) | 0.605 (0.071) | 0.618 (0.047) | 0.631 (0.031) | 0.644 (0.020) | **0.657** (0.013) |
| $\varepsilon_{lumo}$ | $\mu$ | 0.642 (0.053) | 0.656 (0.035) | 0.670 (0.023) | 0.684 (0.015) | 0.698 (0.010) | **0.712** (0.007) |
| $\varepsilon_{lumo}$ | $C_v$ | 0.412 (0.070) | 0.421 (0.046) | 0.430 (0.030) | 0.439 (0.020) | 0.448 (0.013) | **0.457** (0.009) |
| $\mu$ | $C_v$ | 0.824 (0.119) | 0.843 (0.078) | 0.861 (0.052) | 0.879 (0.034) | 0.897 (0.022) | **0.915** (0.015) |

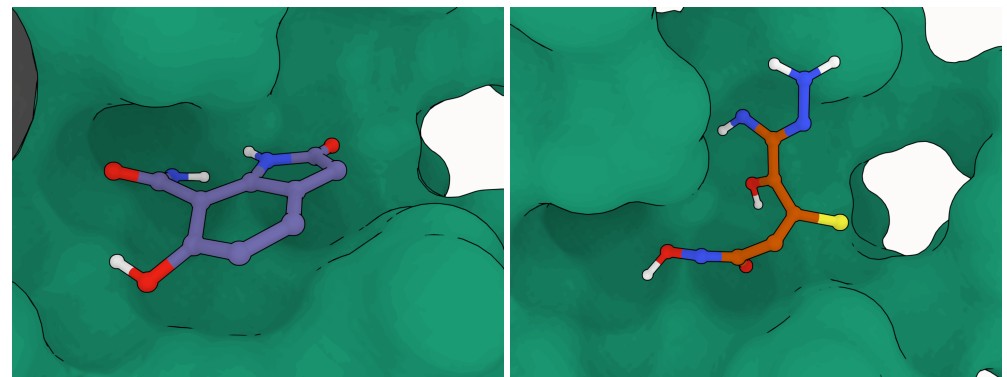

Figure 10: The 1DJY protein ligand generated using DEMO+GeoLDM, where the left side only considers the docking situation and PBValid (vina=-6.3), while the right side additionally considers the case without a ring structure (vina=-5.2).

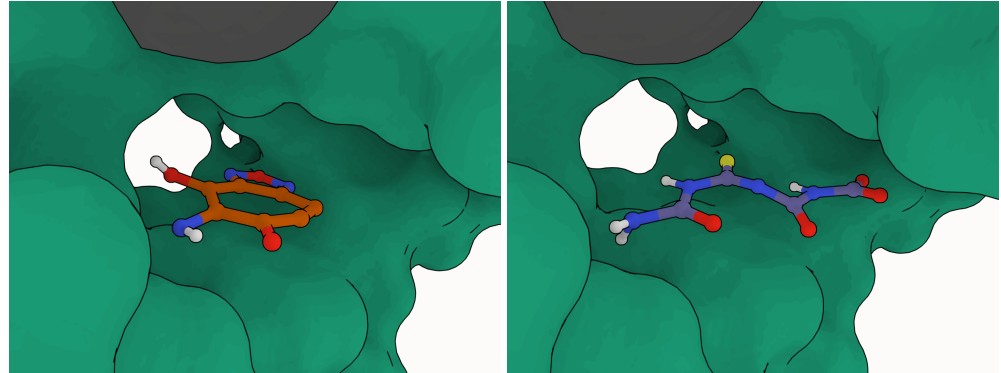

Figure 11: The 2E24 protein ligand generated using DEMO+GeoLDM, where the left side only considers the docking situation and PBValid (vina=-6.4), while the right side additionally considers the case without a ring structure (vina=-5.4).

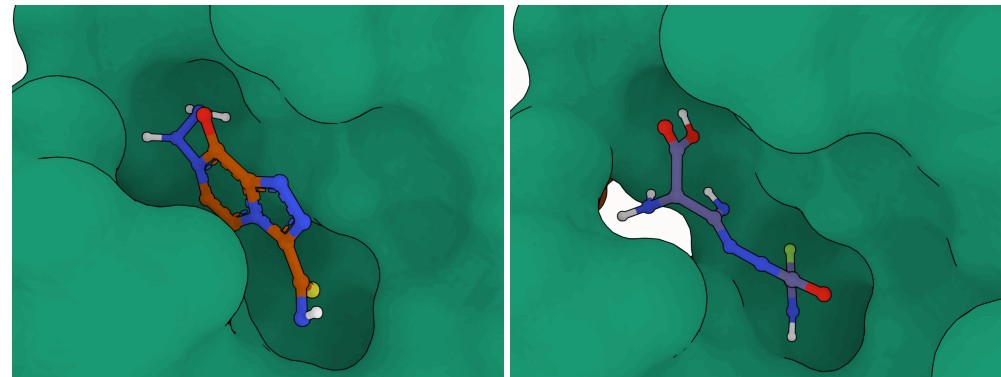

Figure 12: The 3KC1 protein ligand generated using DEMO+GeoLDM, where the left side only considers the docking situation and PBValid (vina=-6.8), while the right side additionally considers the case without a ring structure (vina=-6.8).

Table 7: PoseBusters Metrics in 'dock' Mode

| | |
|---|---|
| **1. Molecule Loading and Structure Validation** | |
| mol_pred_loaded | Whether the molecule was successfully loaded by the prediction engine. |
| mol_cond_loaded | Whether the molecule passed conditional loading and preprocessing. |
| sanitization | Indicates if molecule passed basic chemical sanitization (e.g., valence). |
| inchi_convertible | Whether the structure can be converted to InChI; failure implies severe structural problems. |
| all_atoms_connected | Checks if all atoms are part of a single connected structure. |
| **2. Geometric and Chemical Quality** | |
| bond_lengths | Whether abnormal bond lengths are detected (too long or too short). |
| bond_angles | Checks for abnormal bond angles that may indicate strain. |
| internal_steric_clash | Whether atoms within the same ligand clash with each other. |
| aromatic_ring_flatness | Tests if aromatic rings maintain expected planarity. |
| double_bond_flatness | Verifies flat geometry of double bonds, especially in conjugated systems. |
| internal_energy | Approximate internal energy of the ligand; high values may imply poor conformation. |
| **3. Protein–Ligand Interaction Checks** | |
| protein-ligand_maximum_distance | Maximum distance from any ligand atom to the protein, to detect floating poses. |
| minimum_distance_to_protein | Closest distance between ligand and protein atoms. |
| minimum_distance_to_organic_cofactors | Nearest distance between ligand and any organic cofactor. |
| minimum_distance_to_inorganic_cofactors | Closest distance between ligand and inorganic cofactors (e.g., metal ions). |
| minimum_distance_to_waters | Minimum distance to any water molecule. |
| **4. Steric and Volume Overlap Checks** | |
| volume_overlap_with_protein | Volume intersection between ligand and protein; large overlap indicates steric clashes. |
| volume_overlap_with_organic_cofactors | Overlap with organic cofactors (e.g., NADH), indicating possible collisions. |
| volume_overlap_with_inorganic_cofactors | Volume overlap with metal ions or inorganic components. |
| volume_overlap_with_waters | Volume overlap with crystallographic waters. |

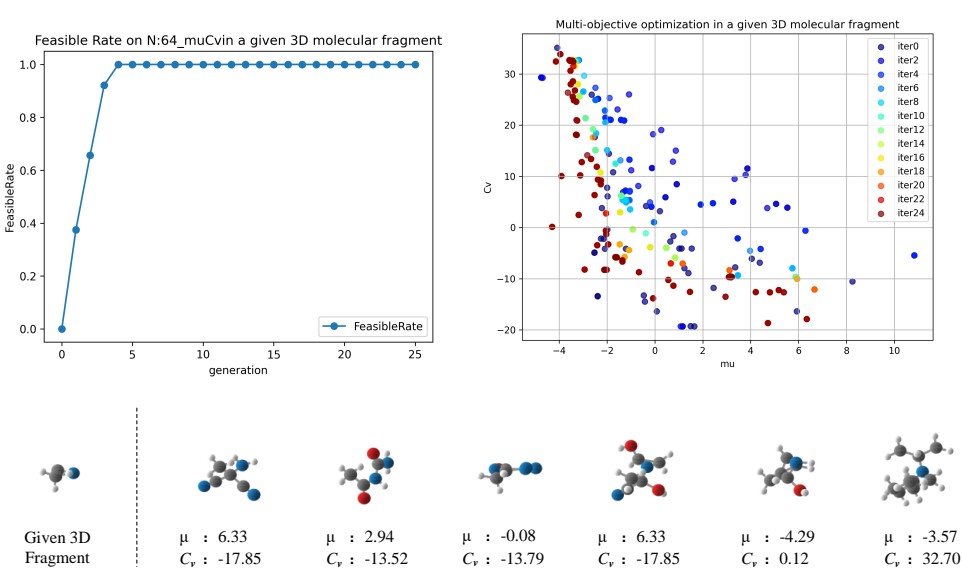

Figure 13: Feasibility and distribution when optimizing $\mu$ and $C_v$ simultaneously and including specific molecular fragments

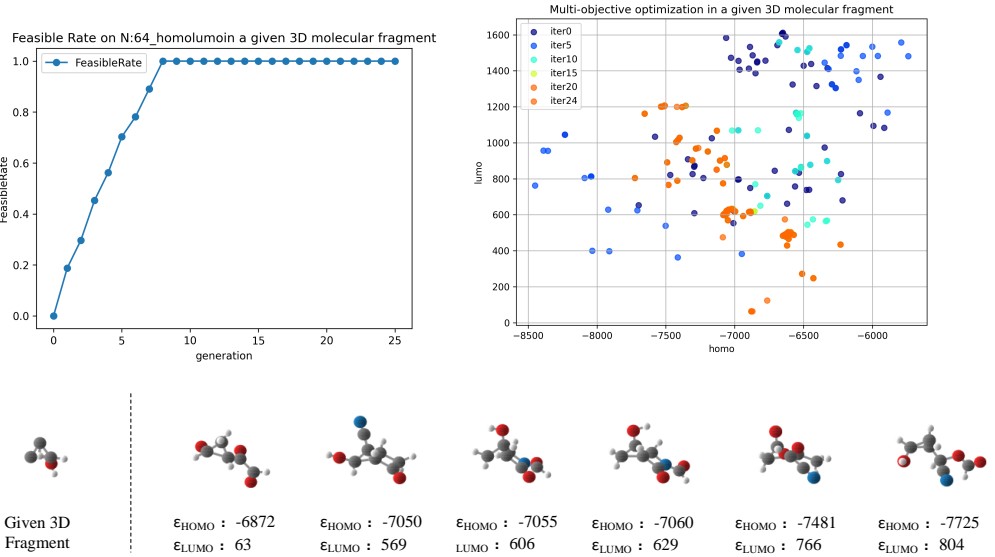

Figure 14: Feasibility and distribution when minimizing $\varepsilon_{homo}$ and $\varepsilon_{lumo}$ simultaneously and including specific molecular fragments

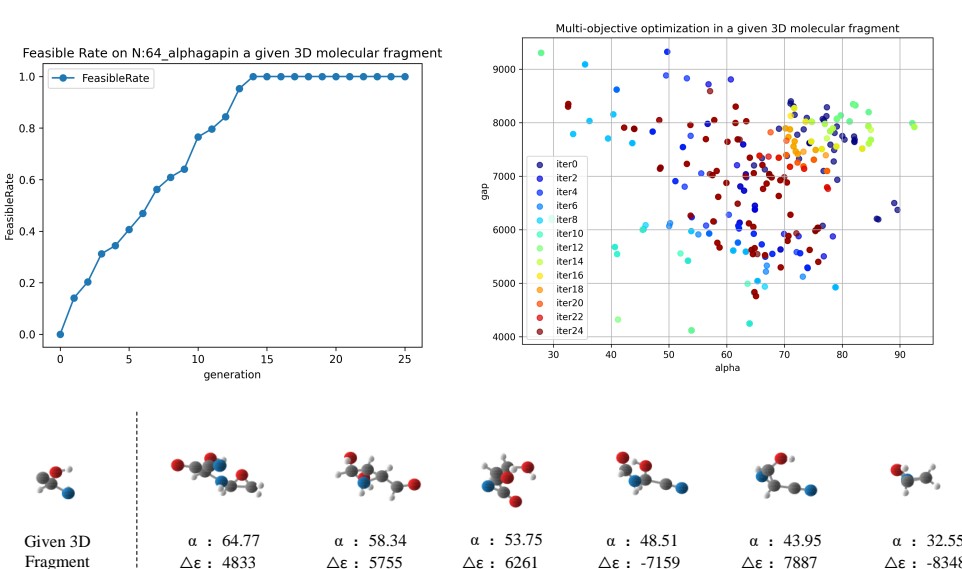

Figure 15: Feasibility and distribution when optimizing $\alpha$ and $\Delta\varepsilon$ simultaneously and including specific molecular fragments

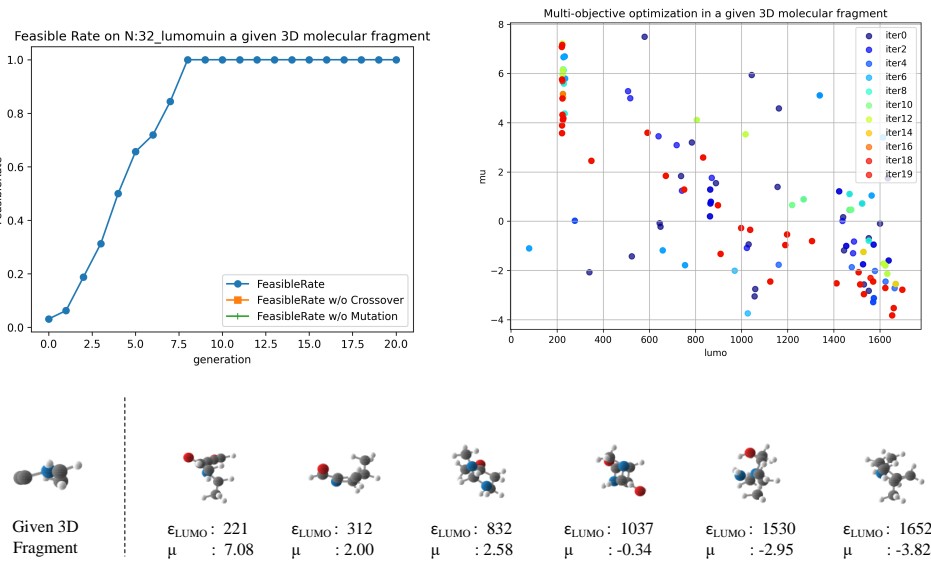

Figure 16: Feasibility and distribution when optimizing $\varepsilon_{lumo}$ and $\mu$ simultaneously and including specific molecular fragments

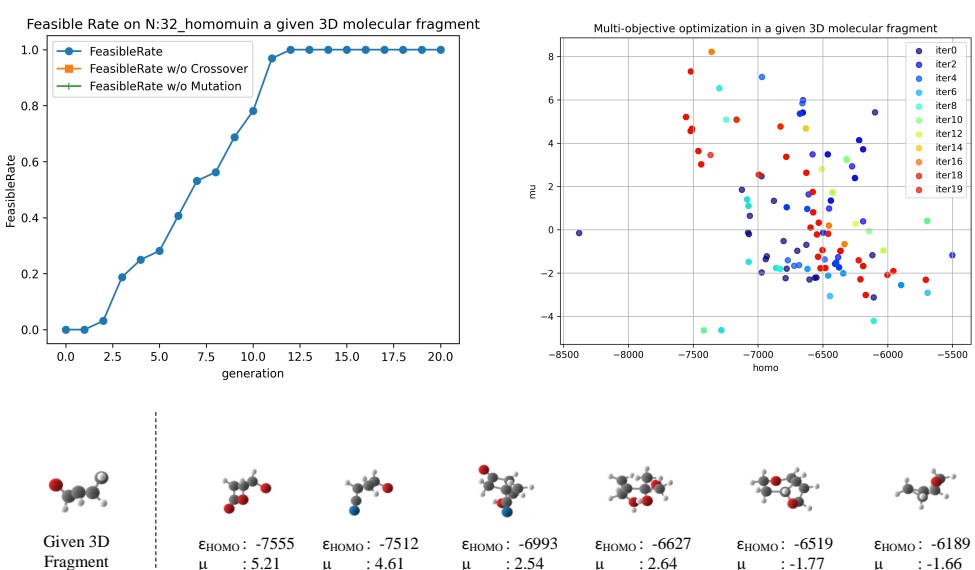

Figure 17: Feasibility and distribution when optimizing $\varepsilon_{homo}$ and $\mu$ simultaneously and including specific molecular fragments

