# OpenReview forum: "DEMO:Diffusion-based Evolutionary Optimization for 3D Multi-Objective Molecular Generation"
_ICLR.cc/2026/Conference — Submitted to ICLR 2026_

### Official Review · Reviewer_3sVG · 2025-10-30

**Soundness:** 3
**Presentation:** 3
**Contribution:** 2
**Rating:** 4
**Confidence:** 4

**Summary:**

The proposed DEMO combines the strengths of evolutionary algorithms (EA), which are effective for multi-objective optimization, with diffusion models capable of generating valid 3D molecules. It efficiently performs 3D molecular multi-objective optimization without requiring any retraining process.

**Strengths:**

- DEMO effectively combines diffusion models with EA, successfully addressing the challenging problem of 3D molecular optimization.
- It utilizes a pretrained diffusion model without any retraining, reducing computational cost and performing multi-objective optimization, which has not been actively explored in previous 3D molecular generation studies.

**Weaknesses:**

- The combination of EA and diffusion models is interesting, but each component is based on well-known techniques, so the overall novelty is limited.
- Although the diffusion model’s forward and denoising processes are well integrated with EA operators, further validation is needed to see whether it works reliably on new objectives or unseen (OOD) samples without retraining.

**Questions:**

- While DEMO’s no-retraining technique is one of its main strengths, wouldn’t the diffusion model still require some additional training when applied to new objectives or molecular datasets?

- Could the DEMO framework be extended to other diffusion models[1] or flow-matching approaches[2]? Although only two baselines are used in this paper, various 3D molecular generation models[1–4] exist. As mentioned in the paper, TFG may be difficult to apply in this setting, but this also seems to highlight a limitation in the model’s extensibility.

- Could you clarify the difference between SOP-MT and SOP-ST? In addition, what distinguishes SOP-MT from MOP-MO? It seems that SOP-MT involves multiple properties that are scalarized into a single objective function — is this interpretation correct? A more detailed explanation would be helpful.

- In Figure 1, do SOP and MOP respectively denote single-objective and multi-objective optimization? If so, the abbreviations may be a bit confusing. Also, placing this figure in the appendix rather than on the main page might be more appropriate.

- Could the proposed DEMO method be applied to optimization problems involving more than three objectives? Since multi-objective optimization typically considers three or more objectives, such an evaluation would help demonstrate the generality of the model. Additionally, including Pareto front visualizations in the discussion section could further aid in interpreting the results.

- In the MOP setting, would it be possible to compare with other fitness design strategies such as Chebyshev scalarization sampling[5] or Dirichlet-weight sampling[6]?

- Is there a particular reason for conducting the ablation study on Crossover (CO) and Mutation (MT) in Table 2? As these are core components of EA, the experiment feels somewhat unconventional. If the goal is to analyze the effects of crossover and mutation within the diffusion latent space, it might be helpful to compare with a naive EA (without diffusion). (Is “TopN” intended to represent such a baseline?)

[1] Le, Tuan, et al. "Navigating the design space of equivariant diffusion-based generative models for de novo 3d molecule generation." arXiv preprint arXiv:2309.17296 (2023).

[2] Lin, Haowei, et al. "TFG-Flow: Training-free Guidance in Multimodal Generative Flow." arXiv preprint arXiv:2501.14216 (2025).

[3] Irwin, Ross, et al. "Efficient 3d molecular generation with flow matching and scale optimal transport." ICML 2024 AI for Science Workshop. 2024.

[4] Irwin, Ross, et al. "SemlaFlow--Efficient 3D Molecular Generation with Latent Attention and Equivariant Flow Matching." arXiv preprint arXiv:2406.07266 (2024).

[5] Chugh, Tinkle. "Scalarizing functions in Bayesian multiobjective optimization." 2020 IEEE Congress on Evolutionary Computation (CEC). IEEE, 2020.

[6] Shin, Dong-Hee, et al. "Offline Model-based Optimization for Real-World Molecular Discovery." Forty-second International Conference on Machine Learning.

---

> ### Author Response · Authors · 2025-11-17
>
> We sincerely thank the reviewer for recognizing the strengths of our work, including its effective combination of EAs and diffusion models and its training-free approach to multi-objective optimization.
>
> **Q1. On the Necessity of Retraining for New Objectives or Datasets**\
> For new *objectives*, DEMO can perform zero-shot optimization by simply swapping the property evaluator. For new *molecular datasets* that are out-of-distribution from the pre-training data, the diffusion model would require fine-tuning or retraining..
>
> In our experiments, we observed that as DEMO converges on the Pareto Front, the proportion of invalid outputs from the diffusion model (e.g., NaN values) increases substantially, indicating that the search has pushed the model to the very edge of its learned distribution.
>
> **Q2. On Extensibility to Other Diffusion or Flow-Matching Models**\
> Yes, DEMO is designed to be model-agnostic. Its core requirement is a model that can function as a projection operator from a noise latent space to the 3D molecular manifold. Therefore, any diffusion or flow-matching model with a clear denoising or reverse-sampling interface can be integrated as a backbone, regardless of its internal architecture (e.g., Transformer, EGNN).
>
> **Q3. On Clarifying SOP-ST, SOP-MT, and MOP-MO**\
> Thank you for this suggestion. We will clarify these terms in the revision:
>
> *   **SOP-ST (Single-Target):** Minimizing the distance to a single target property value.
> *   **SOP-MT (Multi-Target):** Minimizing a scalarized distance to multiple target values in a normalized property space.
> *   **MOP-MO (Multi-Objective):** Finding the Pareto front for multiple objectives without scalarization.
>
> **Q4. On Extending to >3 Objectives and Other Fitness Designs**\
> Yes. Our primary contribution is the 3D molecular evolutionary *operator*, not the selection algorithm itself. This modular design means our operators can be seamlessly integrated into any MOEA framework. This includes MaOEAs (e.g., RVEA, NSGAIII) for handling more than three objectives and decomposition-based MOEAs (like MOEA/D) that use strategies like Chebyshev or Dirichlet-weight scalarization.
>
> **Q5. On the Rationale for the Ablation Study in Table 2**\
> The purpose of this ablation was not to question the necessity of Crossover (CO) and Mutation (MT) for EAs, but to validate that performing these operations **specifically in the noise space** is the key mechanism behind DEMO's success. A "naive EA" without diffusion is not a viable baseline, as it cannot generate valid 3D molecules, which is the very problem DEMO solves (Figure 3). The "TopN" baseline serves as the "no-evolution" comparison, representing a pure sample-and-screen approach.
>
> **Q6. On Overall Novelty**
>
> We agree that DEMO is assembled from well-known building blocks. However, we argue that its novelty lies not in inventing a new primitive, but in designing a novel, systematic, and reproducible framework for practical 3D molecular optimization. In recent work utilizing LLMs for molecular design \[1], the pipeline in practice often reduces to an ad-hoc "prompt engineering + post-processing": careful manual prompt design, repetitive generate-filter-rewrite cycles, and extensive downstream fixes and heuristics. While conceptually similar, DEMO offers greater flexibility as it does not rely on specific, hand-crafted prompt designs.
>
> DEMO can be seen as an evolution of earlier approaches that searched the latent spaces of deep generative models \[2-6]. These methods, which used techniques like BO and EAs on 1D/2D representations, struggled to handle explicit 3D geometric constraints. In contrast, DEMO provides a principled search engine for 3D molecules by: (1) designing evolutionary operators in the **noise/latent space** (rather than operating on symbolic/graph representations), and (2) treating the pre-trained denoiser as a **projection operator** that maps noisy latent recombinations back to the valid chemical manifold.
>
> This combination yields an end-to-end, reproducible optimization loop that (a) operates directly on 3D coordinates, (b) naturally enforces chemical and structural validity via the denoiser, and (c) greatly reduces reliance on manual prompt design. We will clarify and emphasize this methodological contribution in our revision and add these representative citations for prior work.
>
> \[1] Efficient Evolutionary Search Over Chemical Space with Large Language Models. ICLR 2025
>
> \[2] \*Deep Evolutionary Learning for Molecular Design. \*IEEE Computational Intelligence Magazine, 2022
>
> *\[3] Efficient multi-objective molecular optimization in a continuous latent space*. Chemical Science, 2019
>
> \[4]*Evolutionary Multiobjective Molecule Optimization in an Implicit Chemical Space*. JCIM 2024
>
> \[5]Junction Tree Variational Autoencoder for Molecular Graph Generation. ICML.
>
> \[6]Automatic Chemical Design Using a Data-Driven Continuous Representation of Molecules. ACS Central Science

---

### Official Review · Reviewer_MJQd · 2025-10-30

**Soundness:** 2
**Presentation:** 2
**Contribution:** 2
**Rating:** 2
**Confidence:** 4

**Summary:**

This paper presents DEMO (Diffusion-based Evolutionary Molecular Optimization), which is a framework that integrates pretrained 3D diffusion models into evolutionary algorithms for molecular discovery tasks. The method addresses two key issues: (1) evolutionary algorithms in 3D molecular space struggle to generate chemically valid structures due to violation of chemical laws during genetic operations, and (2) diffusion models are inflexible for multi-objective optimization and require expensive retraining when adapting to new properties or constraints. DEMO performs crossover operations in the noise space defined by the diffusion model's forward process by adding Gaussian noise to parent molecules, performing random one-point crossover on the noised representations (both coordinates and features), and then using the pretrained diffusion model's denoising process to restore chemical validity. The noise level is empirically determined through grid search to balance valid crossover against efficient reconstruction. For mutation, the method adds noise to a single parent and denoises it, enabling local exploitation. The framework uses a linear schedule that favors crossover in early generations for global exploration and shifts toward mutation in later generations for local refinement.

**Strengths:**

1. The combination of evolutionary algorithms with pretrained 3D diffusion models for molecular generation is understandable, as it addresses the validity issues inherent in applying genetic operations directly to 3D molecular coordinates.

2. The framework operates in a training-free manner, which can provide practical utility when working with molecular property evaluators that lack gradient information.

3. I appreciate the empirical analysis of noise level selection (t') across different datasets and models.

**Weaknesses:**

**1. Limited Technical Novelty**
- While combining evolutionary algorithms with diffusion models is reasonable from an engineering standpoint, the core contribution offers limited algorithmic innovation. The method essentially applies standard genetic operators (crossover and mutation) to noise-space representations and relies entirely on the pretrained diffusion model's denoising capability.
- In the proposed framework, the crossover operation might be considered merely a linear interpolation in noise space, followed by denoising—a straightforward application of existing techniques rather than a novel methodological advance.
- The individual components (EAs, diffusion models, noise-space operations) are all well-established, and their integration does not introduce new insights.

**2. Questionable Multi-Objective Claims**
- The authors claim to tackle multi-objective optimization, but their experiments only address two-objective problems throughout the paper. Optimizing two objectives is not substantially more difficult than single-objective optimization and does not truly validate the framework's capability for complex multi-objective scenarios.
- For genuine multi-objective optimization validation, the method should handle at least 4 objectives where the complexity of the Pareto front increases dramatically and the trade-offs become significantly more challenging.
-The current two-objective setting raises serious questions about whether this framework is truly beneficial for multi-objective settings or if the claimed advantages would disappear with more realistic problem complexity.

**3. Overstated Pareto Front Discovery Claims**
- In two-objective settings, the Pareto front is relatively simple and easy to approximate. The authors' emphasis on "successfully
and rapidly explores and captures the Pareto front" appears overstated given this limited complexity.
- If the authors want to highlight the method's general performance for 3D molecular generation, they should reduce the emphasis on multi-objective optimization.
- Conversely, if multi-objective optimization is a core contribution, they must demonstrate performance on problems with 4+ objectives where Pareto front approximation is genuinely challenging.

**4. Insufficient Baseline Comparisons**
- The paper lacks comparisons against several relevant baseline approaches. In particular, there appears to be no comparison with state-of-the-art genetic algorithm–based methods with advanced multi-objective optimization techniques, making it unclear how much the diffusion model component actually contributes. If the authors believe they have included sufficient baselines, please clarify.

**5. Unclear Generalization and Applicability**
- The method's performance heavily depends on empirically determined noise levels that vary significantly across datasets. This dataset-dependent hyperparameter tuning requirement raises concerns about the framework's generalizability to new molecular datasets or chemical spaces not covered during training.

- The paper provides no principled method for setting noise level beyond expensive grid search, limiting practical applicability.

**6. Limited Analysis of Component Contributions**
- The ablation studies (w/o CO, w/o MT) show inconsistent results across tasks, with neither crossover nor mutation being uniformly beneficial.  In SOP-MT tasks, removing crossover often improves performance, while in MOP tasks, removing mutation is sometimes better. This inconsistency suggests the framework lacks a coherent design principle and that the components may not synergize as claimed.
- The paper does not adequately explain when and why each operator is beneficial, raising questions about the method's reliability.

**Questions:**

**1. How are multiple objectives scalarized?**
- For SOP-MT tasks (Table 2), the goal is to find molecules that simultaneously match multiple target property values. Equation 7 shows that deviation is calculated as the Euclidean distance. Does this imply a scalarization scheme that assigns approximately equal importance to all property deviations? If so, how are cases handled where certain properties should be prioritized over others? The authors do not clearly discuss or specify how such weighting would be determined.

- Have the authors considered alternative scalarization methods such as Chebyshev scalarization, which can better handle conflicting objectives and avoid bias toward certain regions of the objective space?

**2. Clarification needed on the theoretical justification in Appendix A.2**
- I am confused about the geometric interpretation of “on-manifold” versus “off-manifold” presented in Appendix A.2. The authors claim that the crossover mean is “off-manifold,” while the mutation mean is “on-manifold.” However, both seem to be scaled versions of points that may or may not lie on the molecular manifold. I may be mistaken, but I find this interpretation unclear and would appreciate further clarification.

- If the molecular manifold is defined as the set of chemically valid molecules, then scaling a valid molecule would also produce an "off-manifold" point?

- The theoretical framework relies on the assumption of "Local Fitness Landscape Continuity" (Assumption 1). How do the authors know this assumption holds for molecular fitness landscapes, which can be highly rugged and discontinuous?

**3. Does the "no retraining" advantage truly hold for new objectives and datasets?**
- The authors emphasize that DEMO requires no retraining as a key advantage over existing methods.  However, when applying this framework to entirely new molecular objectives or new chemical datasets,  wouldn't the pretrained diffusion model also require retraining or fine-tuning to properly capture the distribution of valid structures in these new domains?

- The current evaluation uses diffusion models pretrained on QM9 and GEOM-Drugs and tests on the same datasets. This does not validate the "no retraining" claim for genuinely out-of-distribution scenarios. Can the authors demonstrate that their framework works with a diffusion model pretrained on Dataset A but applied to optimize molecules from a completely different Dataset B without any retraining?

**4. Can this approach be applied to 2D molecular generation/optimization?**
- The authors emphasize 3D molecular generation throughout the paper. However, I wonder whether the core idea of performing genetic operations in diffusion noise space could also be applied to 2D molecular generation. There has been substantial research on multi-objective 2D molecular generation and optimization, including benchmarks such as PMO (Practical Molecular Optimization) introduced by Gao et al. at NeurIPS 2022. Considering this, exploring 2D molecular generation would be an important and natural extension. Have the authors considered this direction?

---

> ### Author Response · Authors · 2025-11-18
>
> **1. Limited technical novelty**\
> We agree DEMO builds on existing pieces, but its contribution is a **principled, reproducible integration**: (i) designing evolutionary operators **in diffusion noise space**, and (ii) using the pretrained denoiser as a **projection operator** back to the valid chemical manifold. This combination yields a practical, training-free 3D molecular search engine that couples guaranteed chemical validity with black-box multi-objective optimization — more than mere ad-hoc engineering.
>
> **2. Multi-objective claims (Weakness 2 & 3)**\
> Optimizing two objectives (MOP) is significantly harder than one, as the trade-offs are often unknown and conflicting. While Many-Objective Optimization (MaOP, >3 objectives) is more complex still due to weakened Pareto dominance, our core contribution is a novel **3D molecular evolutionary operator**, not a new multi-objective selection algorithm.  However, MOP is a critical and unavoidable need in molecular discovery, and MOEAs are the primary tool to address it. Therefore, demonstrating our operator's effectiveness in a multi-objective setting is a natural and powerful way to validate its utility. The two are tightly coupled. Our operator is modular and can be readily integrated into established MaOEAs (e.g., NSGA-III, MOEA/D) to solve such problems.
>
> **3. Baseline comparisons**\
> This is an excellent suggestion. **We will add comparisons against more MOEA baselines in the revision.** We note, however, that the performance differences between established selection algorithms like SPEA2, NSGA-II, and MOEA/D are often marginal on many problems.
>
> **4. Dependence on t' and generalization**\
> Determining t' is a key challenge. Ideally, t' should be minimal and pair-specific to ensure any combination of two parents remains on the thickened manifold, but this is intractable. Our pre-defined t' is a pragmatic heuristic, and the cost of finding it via a simple search is low. A more advanced approach, which we will add as an option in our code, is to use an adaptive schedule that starts with a high t' and anneals it until validity begins to drop.
>
> **5. Component ablations and variability**\
> The varying ablation results reflect landscape-dependent needs: MOP tasks benefit more from global exploration (crossover), while SOP-MT tasks often need local refinement (mutation). This variability shows framework flexibility rather than lack of coherence; we will add a targeted discussion explaining when each operator is expected to help.
>
> ***
>
> ### Answers to the reviewer’s specific questions
>
> **Q1. How are multiple objectives scalarized?**\
> For SOP-MT we normalize each property (min/max over current population) and use an equal-weight Euclidean distance as the deviation score. The framework is fully compatible with alternative scalarizations (e.g., MOEA/D decompostion with Chebyshev) and with task-specific weighting — we will clarify this and note these options in the revision.
>
> **Q2. Clarification of “on-manifold” vs “off-manifold” (Appendix A.2)**
>
> *   Intuition: **Mutation** perturbs a single valid molecule; its noised mean remains close to the original manifold point.
> *   **Crossover** stitches parts of two different molecules, producing a chimeric mean that typically violates chemical constraints and is therefore more “off-manifold.”
> *   Scaling a valid structure (or other coordinate distortions) also generally breaks precise inter-atomic relations and moves the point off the valid manifold.\
>     We will rewrite Appendix A.2 for clarity, make the geometric claims more precise, and explicitly note exceptions (e.g., activity cliffs). We will also cite matched-molecular-pair (MMP) evidence to support practical local continuity while acknowledging empirical counterexamples.
>
> **Q3. Does “no retraining” hold for new objectives/datasets?**\
> To be precise: our "no retraining" claim applies to **new objectives**, where we only need a new evaluator. For entirely new chemical spaces (OOD datasets) where the pre-trained model's prior is a poor fit, fine-tuning or retraining would be necessary. This stance is consistent with related training-free guidance works like TFG and MUDM.
>
>
> **Q4. Applicability to 2D molecular generation?**\
> Yes. DEMO’s operator is modality-agnostic: any diffusion model with a forward noise process and reverse denoiser (including 2D graph or SMILES diffusion models) can host the same noise-space genetic operators.

---

### Official Review · Reviewer_vcKG · 2025-10-31

**Soundness:** 3
**Presentation:** 3
**Contribution:** 3
**Rating:** 4
**Confidence:** 3

**Summary:**

This work has proposed a 3D single and multi-objective molecular evolutionary algorithm that leverages a pretrained diffusion model. DEMO integrates diffusion models into an evolutionary loop to improve validity and accelerate search. This also combined 3D diffusion backbone (EDM, …) with classical operators executed in noise space. The framework is also training-free and demonstrates improvements across multiple tasks.

**Strengths:**

To my knowledge this is the first framework that combines evolutionary algorithms with 3D diffusion models, which is novel and interesting.
The problem formulation is clear and grounded in the goals of multi objective molecular design.
The effectiveness of the framework is shown across four tasks using the same pretrained model

**Weaknesses:**

More detailed explanations of the reported metrics in the tables, including Table 1 and 2, are needed.

The approach is interesting and reaches state of the art on some metrics, yet results are not fully consistent, as different diffusion backbones share or swap the best scores across metrics.

In the Tasks and Datasets section, please specify the exact objectives for SOP and MOP, for example which properties are optimized and how they are computed.

Performance of crossover and mutation depends strongly on the noise level t prime, chosen by empirical tuning. The authors mentioned that this as balancing manifold linearity for valid crossover against information retention for reconstruction. This task dependent tuning slightly weakens the plug and play story.

**Questions:**

The authors mentioned that the 3D crossover is a simple one-point split on the atom sequence, leading to a “random partitioning”. Given the arbitrary nature of the atom list, how can this unguided fragmentation effectively combine meaningful molecular fragments to do chemically valid exploration that is superior to a well-designed chemically-aware crossover operator?

The noise level t' is reduced linearly across generations (Algorithm 1, line 4). Did the authors explore a dynamic, fitness-dependent t' scheduling, where the noise level is adapted based on the current population's fitness, rather than a fixed linear decrease?

---

> ### Author Response · Authors · 2025-11-16
>
> We sincerely thank the reviewer for their positive evaluation of our work. Below are our responses to your questions.
>
> **1. On Objectives and Metrics in Tables 1 & 2**\
> For Single-Objective Optimization (SOP), we optimize single QM9 properties (e.g., HOMO, ∆ε) to match a target value, evaluated using Mean AbsoluteError (MAE). For Multi-Objective Optimization (MOP), we co-optimize property pairs (e.g., minimize ∆ε and maximize α) to find the Pareto front, evaluated using Hypervolume (HV), as detailed in Section 3.2. We will clarify this in the main text.
>
> **2. On the Inconsistency of Results Across Backbones**\
> This is expected because DEMO searches the distribution learned by its backbone. Since EDM and GeoLDM learn different distributions due to their distinct architectures, the optimal molecules they can generate will naturally vary, leading to different performance on specific tasks.
>
> **3. On the Effectiveness of Random Crossover in Noise Space**\
> Our crossover is effective because the crossover is on noisy samples, the noising process preserves high-level molecular structure while relaxing low-level constraints (e.g., bond angles). This allows for swapping meaningful structural features in the noise space, which the denoiser then reconstructs into valid molecules. The evolutionary algorithm's selection pressure amplifies this by preserving molecules with superior features, thus focusing the search on high-fitness regions of the chemical space.
>
> **4. On the Choice and Scheduling of Noise Level t'**\
> You are correct that t' controls the critical exploration-exploitation trade-off. In the revision, we will add a discussion on adaptive t' scheduling as a compelling direction for future work. Theoretically, the ideal t' is pair-specific—just enough noise to ensure any combination of two parents remains in a recoverable region of the manifold. Since computing this is intractable, our linear schedule, based on a low-cost pre-exploration, offers a practical and effective heuristic.
>
> Furthermore, the best approach is to set a relatively large t' (e.g., 400) and allow it to decrease continuously during the evolutionary process, while observing the reasonableness of the generated samples. When the reasonableness significantly decreases, we choose not to decrease t' further. However, in the paper, we pre-define a t'. This helps us present to readers what the optimal value of t' is, and we will add this option to the code.

---

### Official Review · Reviewer_DBv8 · 2025-10-31

**Soundness:** 3
**Presentation:** 2
**Contribution:** 2
**Rating:** 2
**Confidence:** 5

**Summary:**

The paper proposes to replace the mutation and crossover operations in evolutionary algorithms with diffusion-based denoising, enabling property-guided molecular generation. The method is evaluated on small-molecule datasets, such as QM9 and GEOM-DRUG datasets, and on a protein–ligand optimization dataset where Vina scores are used as the oracle.

**Strengths:**

Overall, the manuscript is well-structured and reads smoothly, making it relatively easy to follow the proposed methodology.

**Weaknesses:**

## Experimental Design
The paper attempts to optimize certain QM9 properties as the objective, but the motivation for those is unclear. QM9 is originally designed as a regression benchmark, and I do not see a strong justification for treating these properties as optimization targets. It is also unclear how the reference scores or target ranges for these QM9 objectives were determined.

In contrast, the Vina score–based optimization task appears to be much more meaningful and realistic for molecular design. In fact, I would argue that this should have been the main experimental setting, rather than QM9.

## Multi-objective Setting
For multi-objective optimization, the more standard and practically relevant setup involves objectives such as Vina score + QED + SA, as used in recent works such as Saturn [1] and fRAG [2]. I believe adopting such a setting would provide a stronger and more convincing evaluation of the proposed method.

## Number Objectives
The paper considers the optimization of only two objectives for the multi-objective experiments, yet refers to this as the Pareto front and multi-objective optimization, and this setup feels somewhat limited. Recent studies typically explore 4–6 objectives simultaneously to better reflect the complexity of real-world molecular design [3]. Extending the evaluation to higher-dimensional objective spaces would make the results more compelling.


[1] Saturn: Sample-efficient Generative Molecular Design using Memory Manipulation (Arxiv 2024)

[2] Drug Discovery with Dynamic Goal-aware Fragments (ICML 2024)

[3] Efficient Evolutionary Search Over Chemical Space with Large Language Models (ICLR 2025)

**Questions:**

A recent ICML 2025 paper [4] pretrained neural network-based operators for crossover, which seems conceptually related to DEMO. However, that work explicitly pretrains the neural network to learn the crossover operator, while it is unclear whether DEMO applies any GA-specific training to align the diffusion model with the evolutionary algorithm. It would be helpful if the authors could clarify how the diffusion model was trained or adapted to effectively serve as a mutation/crossover operator within the EA framework.

[4] Offline Model-based Optimization for Real-World Molecular Discovery (ICML 2025)

---

> ### Author Response · Authors · 2025-11-16
>
> We thank the reviewer for their constructive feedback and address their points below.
>
> **1. On the motivation for optimizing QM9 properties:**
>
> We respectfully disagree that optimizing QM9 properties lacks motivation. While QM9 is a regression benchmark, its fundamental quantum-mechanical properties are directly relevant to real-world material design, making it a valid and challenging optimization testbed. For instance, in organic photovoltaics (OPVs), a key goal is to minimize the HOMO-LUMO gap (∆ε) to increase visible light absorption while maximizing polarizability (α) to improve the light absorption cross-section. Similarly, for organic semiconductors, one must tune properties like ∆ε, μ, and α to control charge mobility. Crucially, these diverse properties must be optimized simultaneously, yet the relationships between them are often conflicting and unknown. This creates a complex optimization landscape that is difficult for many methods to navigate. This is precisely the challenge that DEMO, with its population-based, multi-objective framework, is designed to solve. It excels at exploring these intricate trade-offs to discover a set of optimal solutions (the Pareto front). Therefore, our experiments on QM9 are not only well-grounded but also serve as a challenging testbed that highlights DEMO’s core strengths.
>
> **2. On reference scores and target ranges:**
>
> We clarify that our methodology is detailed in the appendix. To prevent data leakage, we split the QM9 dataset for training the diffusion backbone and property predictors separately (A.3.1). For a fair evaluation, Hypervolume (HV) bounds are normalized based on the property range of the training set (A.3.2).
>
> **3. On the number of objectives:**
>
> Our primary contribution is a novel evolutionary *operator* for 3D molecular optimization, not a new selection algorithm. While we validate our operator in a standard Multi-Objective Optimization (MOP) setting (2-3 objectives), it is modular and can be readily integrated into established Many-Objective EAs (4 and more objectives, e.g., NSGA-III, MOEA/D) to handle a higher number of objectives.
>
> **4. On GA-specific training for the diffusion model:**
>
> Thank you for the reviewer's reminder. DEMO did not perform any pre-training or joint training of the diffusion model for GA; we directly used the offline pre-trained 3D diffusion model as a fixed denoising/projection module, performed crossover/mutation on the parent in the noise space, and then the denoiser mapped it back to the chemically feasible conformation, thereby performing efficient search on the probability distribution learned by the model. Reference \[4], on the other hand, uses explicit learning of neural crossover operators to align recombination strategies. Both approaches have their advantages: DEMO has lower engineering overhead and is naturally compatible with black boxes and multi-objective targets; while the operator-learning method may achieve higher task alignment when there is sufficient training data and precise control of fragment insertion/exclusion is required.
>
> We also note an important modality difference: [4] operates on 2D molecular representations, whereas DEMO targets 3D molecules. Generating valid 3D structures is substantially more demanding—geometry, bond lengths/angles, stereochemistry, and nonbonded contacts must all be respected—so naive fragment stitching that may be acceptable in 2D often produces invalid or highly strained 3D geometries. Addressing this in a learned 3D stitching framework would require additional geometry-aware alignment and relaxation mechanisms, which is why we favor using a robust 3D denoiser as a projection operator in DEMO. We will add a short discussion of these modality differences and their implications in the revised manuscript.
>
> Li, B., et al. (2015). Many-Objective Evolutionary Algorithms: A Survey. *IEEE Transactions on Evolutionary Computation*.\
> Deb, K., & Jain, H. (2013). An Evolutionary Many-Objective Optimization Algorithm Using Reference-Point-Based Nondominated Sorting Approach... *IEEE Transactions on Evolutionary Computation*.\
> Zhang, Q., & Li, H. (2007). MOEA/D: A Multiobjective Evolutionary Algorithm Based on Decomposition. *IEEE Transactions on Evolutionary Computation*.

---

### Meta-Review · Area_Chair_Q7X3 · 2026-01-09

**Summary:**

This paper proposes DEMO, which combines pretrained 3D diffusion models with evolutionary algorithms (EA) for multi-objective molecular optimization.

The paper presents a training-free algorithm that integrates EA and diffusion models. The main benefits are improved chemical validity of generated structures and better optimization for multi objectives.

However, there are several main concerns:

- Limited conceptual and methodological novelty. The core idea of using a pretrained diffusion model within an EA is a natural and incremental combination of two well-established approaches. The paper does not clearly articulate what new insight arises from this combination.

- Insufficient evaluation of multi-objective optimization.
Although the method is positioned as a multi-objective optimizer, the experiments consider only two objectives, whereas prior work commonly considers 4-6 objectives. During the rebuttal, the authors clarified that the method can be applied to more than two objectives; however, without empirical evidence, this claim is difficult to assess.

- Lack of recent baselines.
The authors indicate that recent baselines will be added in a revision, but no new results are presented in the current version.

Overall, while the paper explores a promising combination of EA and diffusion models, its empirical benefits are not clearly demonstrated, and the algorithmic insight is limited.

**Reviewer Concerns:**

Addressed:

- Motivation of optimizing QM9 properties.

- Comparison to a recent ICML2025  paper.

Unaddressed:

- Number of objectives.
- Limited technical novelty.
- Lack of recent baselines.
- Common multi objective setup is not tested
- Empirical results are inconsistent and thus raise questions about the effect of the proposed components.

**Reviewer Scores:**

They are likely to remain similar.

---

### Decision · Program_Chairs · 2026-01-26

Reject